# Rubisco forms a lattice inside alpha-carboxysomes

Lauren Ann Metskas [1,2,3] ✉, Davi Ortega[1], Luke M. Oltrogge [4], Cecilia Blikstad [4,6], Derik R. Lovejoy[2], Thomas G. Laughlin [4,7], David F. Savage [4] & Grant J. Jensen [1,5] ✉

Despite the importance of microcompartments in prokaryotic biology and bioengineering, structural heterogeneity has prevented a complete understanding of their architecture, ultrastructure, and spatial organization. Here, we employ cryo-electron tomography to image α-carboxysomes, a pseudo-icosahedral microcompartment responsible for carbon fixation. We have solved a high-resolution subtomogram average of the Rubisco cargo inside the carboxysome, and determined the arrangement of the enzyme. We find that the *H. neapolitanus* Rubisco polymerizes in vivo, mediated by the small Rubisco subunit. These fibrils can further pack to form a lattice with six-fold pseudo-symmetry. This arrangement preserves freedom of motion and accessibility around the Rubisco active site and the binding sites for two other carboxysome proteins, CsoSCA (a carbonic anhydrase) and the disordered CsoS2, even at Rubisco concentrations exceeding 800 μM. This characterization of Rubisco cargo inside the α-carboxysome provides insight into the balance between order and disorder in microcompartment organization.

Many prokaryotes employ compartmentalization to sequester and facilitate biochemical activities. One example of this organization is the bacterial microcompartment, a collection of enzymes enclosed in a proteinaceous shell that selectively restricts passage of key intermediates and improves on-target catalysis[1,2]. Interest in microcompartment bioengineering has recently grown, particularly for transplanting or reconstructing carbon fixing microcompartments to non-native hosts[3–7]. However, while reconstituted assemblies of shell proteins and other components have been studied at high resolution[3,8–10], the supramolecular organization of the microcompartment interior remains elusive.

The α-carboxysome (CB) is a microcompartment responsible for carbon fixation in many cyanobacteria and chemoautotrophs[11]. It contains two enzymes, Rubisco and carbonic anhydrase, encapsulated within a pseudo-icosahedral protein shell (Fig. 1)[12]. A third abundant

component, CsoS2, is a disordered scaffold protein that binds both the shell and Rubisco and is responsible for Rubisco's encapsulation[13–16]. CBs can also contain lower-abundance proteins, such as a Rubisco activase[17].

Rubisco is often considered a poor enzyme, with slow turnover and an undesirable off-pathway reaction with oxygen[18]. To circumvent these problems, some autotrophic bacteria employ a $CO_2$ concentrating mechanism (CCM)[19,20]. In the CCM, bicarbonate is actively pumped into the cytosol and then diffuses through the semi-permeable protein shell into the carboxysome where carbonic anhydrase converts it to $CO_2$. In this way, high local $CO_2$ concentrations are provided to the encapsulated Rubisco, maximizing its turnover and outcompeting oxygenase activity. Although this simple model explains how increased on-target turnover can be achieved, recent work on CBs suggests there may be other important structural

[1]Division of Biology and Biological Engineering, California Institute of Technology, Pasadena, CA, USA. [2]Biological Sciences Department, Purdue University, West Lafayette, IN, USA. [3]Chemistry Department, Purdue University, West Lafayette, IN, USA. [4]Department of Molecular and Cell Biology, University of California, Berkeley, CA, USA. [5]Department of Chemistry and Biochemistry, Brigham Young University, Provo, UT, USA. [6]Present address: Department of Chemistry, Ångström Laboratory, Uppsala University, Uppsala, Sweden. [7]Present address: Division of Biological Sciences, University of California, San Diego, CA, USA. ✉e-mail: metskas@purdue.edu; grant_jensen@byu.edu

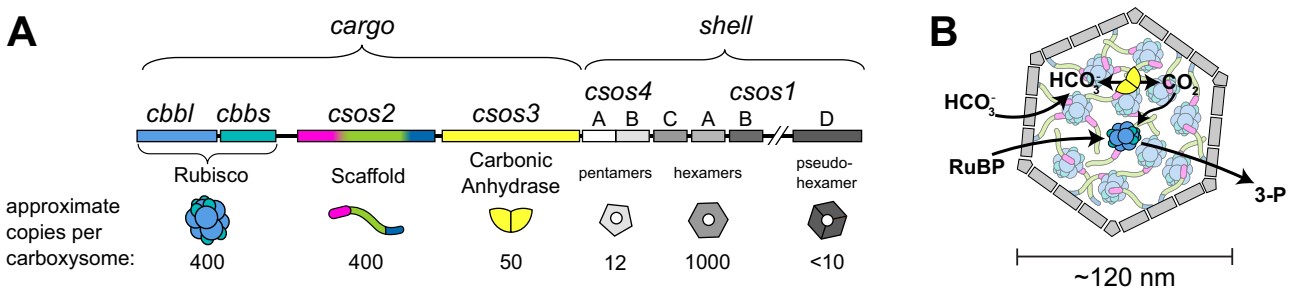

**Fig. 1 | Schematics of the *H. neapolitanus* α-carboxysome. A** The *H. neapolitanus* CB operon, with estimates of relative protein copy numbers per CB[11, 17]. **B** Current working model of a carboxysome. Carbonic anhydrase converts bicarbonate to carbon dioxide; Rubisco then uses the carbon dioxide to convert one molecule of ribulose-1,5-biphosphate to two 3-phosphoglycerate molecules.

mechanisms; for example, carbonic anhydrase activity is likely regulated in a post-translational fashion[3,10,21]. It remains unknown how the ultrastructural features of the CB, such as the inherent protein-protein interactions driving assembly, may alter enzyme activity through enzymatic scaffolding or other regulation.

Here, we present a high-resolution view of Rubisco structure and organization within the *Halothiobacillus neapolitanus* α-carboxysome. We discover that Rubisco can arrange into a low-periodicity lattice inside the CB at high concentration, driven by polymerization of Rubisco itself. This feature of the CB may facilitate high-density encapsulation of Rubisco without compromising enzyme activity.

## Results

We collected cryo-electron tomograms of 139 purified CBs displaying a range of sizes and Rubisco concentrations (Fig. 2A, Supplementary Fig. 1). Using a customized particle-picking approach designed to pick every Rubisco (Methods), we identified 32,930 Rubisco particles and carried out subtomogram alignment and averaging to elucidate the structure of Rubisco inside the CB.

Our 4.5 Å Rubisco map shows strong agreement with crystal structure 1SVD of the same enzyme (Fig. 2B). Helices align well to predicted locations in the crystal structure, and bulky side chain densities often match crystal structure positions (Fig. 2C). However, we observe a modest discrepancy between our density map and the crystal structure in the small Rubisco subunit. If the large and small Rubisco subunits are docked as rigid bodies, this results in a slight (1 Å displacement) tilt of the beta-turn-beta feature of the small subunit (Y96 and adjacent residues, Supplementary Fig. 2). This slightly widens the recently identified binding pocket for carboxysomal proteins CsoS2 and CsoSCA[13,22], but leaves the top and bottom interfaces of the complex unchanged relative to the crystal structure.

The purified CBs display a range of Rubisco concentrations (Supplementary Fig. 1, Supplementary Table 1). Nearly half the CBs displayed ordered packing, with high concentrations of Rubisco packed tightly into aligned rows (Fig. 2A right; Supplementary Fig. 6B). Other dense CBs contain similar concentrations of Rubisco but lack the ordered packing, instead showing shorter Rubisco fibrils (chains of complexes), isolated fibrils, or single complexes without strong alignment (Fig. 2A left; Supplementary Fig. 6C). Finally, roughly a third of CBs contain lower concentrations of Rubisco (Fig. 2A center, Supplementary Fig. 6D; sparse CBs may be over-represented in purified CBs due to purification effects). A nearest-neighbor Rubisco alignment search across all CBs revealed a concentration dependence: Rubisco orientation is random at low concentration but becomes increasingly non-random as concentration rises (Supplementary Fig. 3).

All ordered CBs contain both nematic and isotropic phases in different areas of the interior. The short fibril lengths and disorder in the CB preclude traditional analyses for lattices and liquid crystals, but can be characterized through spatial analysis (Methods). Rendering all

Rubisco fibrils within a CB displays the ordered phase: a loose hexagonal lattice-like ultrastructure of Rubisco fibrils twisting about each other with six-fold pseudo-symmetry (Fig. 3B, Supplementary Movie 1). To identify CBs containing a large twisted lattice, we used the second rank order tensor to search for a single shared axis among all Rubiscos within each CB (Fig. 3A, Methods)[23]. Its scalar, S, quantifies the alignment of all Rubiscos inside a CB, with higher S values (S > 0.35) indicating predominance of a single shared axis (Fig. 3B, Supplementary Movie 1). CBs with a large central lattice are only found at Rubisco concentrations above 650 μM (Fig. 3A). The fibrils have lateral Rubisco distances of 12.5 ± 0.7 nm with a tilt of 10 ± 3 degrees (roughly 1 nm space between Rubisco edges, Fig. 3D).

Despite this order, no rigid scaffold is observed holding the Rubisco lattice together. An average of adjacent fibrils shows faint densities between the large and small subunits of adjacent Rubiscos (Fig. 3D), suggesting a flexible, sub-stoichiometric binding partner may be present in the lattice. CsoS2 and carbonic anhydrase are both known to bind Rubisco through low-affinity interactions with disordered peptides[22]. CsoS2, in particular, has the appropriate disorder, length and Rubisco binding sites to serve this role[13]. The Rubisco termini have also been suggested to participate in intermolecular interaction, which could provide an alternative route for assembly[24].

The fibrils forming the lattice appear to be the product of Rubisco polymerization. The ratio of bound to free Rubisco scales with total Rubisco concentration in the CB (Fig. 4A), and a difference map of Rubisco monomer removed from a fibril average shows no additional density except for the other Rubiscos (Fig. 4C). Polymerization is observed in all packing types, though fibril lengths are longer inside the Rubisco lattices of ordered CBs (Fig. 3C, Supplementary Fig. 6B–D). This is likely the effect of oriented macromolecular crowding restricting diffusion and encouraging rebinding if dissociation occurs.

The Rubisco-Rubisco interaction appears to be low affinity, as the four binding sites on the C4 planar surface are not fully bound at any time. The purified CBs show a bending angle of 3 ± 2.5 degrees and a 14-degree standard deviation in the twist between stacked Rubiscos (Fig. 4C), and non-symmetrized subtomogram averages of the fibrils have incomplete occupancy across the sites. It is likely that the low affinity of the Rubisco-Rubisco interaction results from a fast $k_{off}$ rate at each site, decreasing the likelihood of all sites being bound at any one time and resulting in the wobbly interaction observed. This is consistent with the high concentrations associated with fibril formation in Figs. 3A and 4A.

We identified the Rubisco-Rubisco interaction site by performing subtomogram alignment and averaging of Rubisco within fibrils, using a tight mask at the upper interface to counteract the non-zero bending angle (Fig. 4B). The strongest density is between the tip of small subunit helix 1 on one Rubisco and the center of small subunit helix 3 of

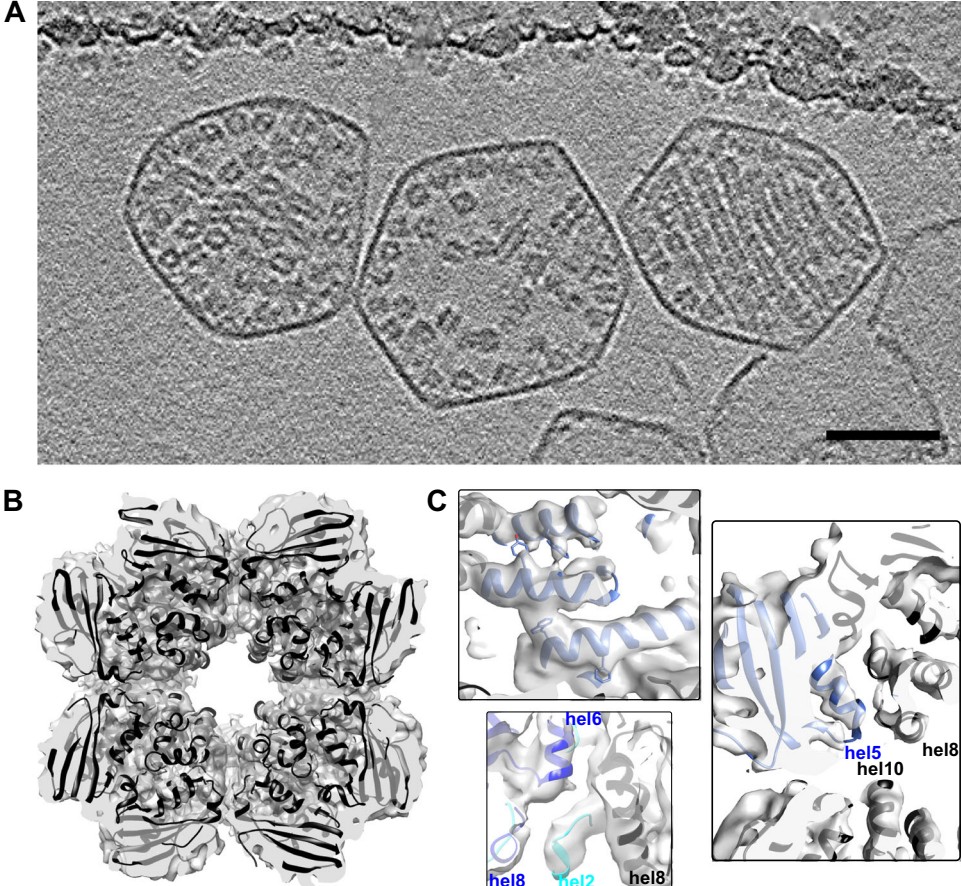

**Fig. 2 | Rubisco subtomogram averaging. A** 4.4 nm-thick orthoslice showing three carboxysomes with different packing behaviors. From left: dense, sparse, ordered. Scale bar 50 nm. **B** Slab view of the 4.5 Å density map, with the 1SVD crystal structure docked inside (ribbon). **C** 1SVD crystal structure docked within the density map (blue, large subunit; cyan, small subunit; black, neighboring subunits). Top: some bulky side chains (Y244, W276, W207, F213) are visible in the map. Right: helices are resolved and the docked crystal structure fits well across three large subunits. Bottom: the small subunit docks well between two large subunits.

the binding partner, though the resolution is too poor to resolve the side chains driving the interaction. The variability in fibril parameters suggests that multiple loose interactions are likely responsible for the binding, consistent with the poor resolution in the Rubisco-Rubisco interface.

Previously solved crystal structures of this Rubisco also display variable interactions in this region, underscoring the lack of a single, high affinity interaction at this site. *H. neapolitanus* Rubisco structures 1SVD (Rubisco) and 6UEW (Rubisco-CsoS2 complex) both show alternative longitudinal interactions in the crystal structure involving helix 3 (Supplementary Fig. 4). Both crystal structure interactions lie within the range represented in our data, suggesting these conformations may be sampled in vivo (though not the dominant conformation). However, the side chains observed in the interaction site of the crystal structures or our interface model have neither obvious complementarity for interaction nor evolutionary conservation (Supplementary Figs. 4 and 5). Taken together, these observations suggest that Rubisco packing at the concentrations reached in both protein crystals and CBs can promote low-affinity interactions with a range of symmetries and periodicities.

The Rubiscos adjacent to the shell occasionally participate in the long fibrils, but not consistently, and appear to behave differently than Rubisco in the CB interior. The number of Rubiscos in this shell-proximal layer scales with CB volume (Fig. 5A), and the layer is well-populated even in otherwise sparse CBs. We analyzed the angle of the Rubisco C4 axis relative to the shell and found a predominantly random distribution with a skew toward perpendicular orientations (likely reflecting the participation in fibrils, which would favor this orientation) (Fig. 5B).

Surprisingly, we were unable to identify a physical linkage between Rubisco and the shell. We compared a subtomogram average of shell-adjacent Rubisco with an average of interior non-fibril Rubisco and found no well-defined density at the binding site for CsoS2 (Fig. 5C). We did, however, observe many shell-attached densities in the tomograms, which are best visible in tomograms of broken CB shells (Fig. 5D). Some of these densities are the correct size for the carbonic anhydrase CsoSCA, which previously has been suggested to be shell associated[25]. However, recent in vitro studies show that CsoSCA binds Rubisco but has no detectable interaction with shell proteins, suggesting that these densities—which could not be assigned unambiguously—could alternatively be shell-attached Rubisco activase complexes or aggregates of CsoS2[17,22].

The most consistent feature of the CBs is their compositional and structural heterogeneity. Because purification can impose constraints in size and density, we collected tomograms of CBs inside intact *H. neapolitanus* cells (Fig. 6A). Both the dense and ordered CB morphologies are visible in vivo (Fig. 6B–C, Supplementary Fig. 6E), along with additional morphologies lost during purification such as elongated structures containing layers of non-filamentous Rubisco (Fig. 6D, Supplementary Table 1). In vitro and in vivo, subpopulations of CBs occasionally contain what appears to be an aggregate of polypeptide, presumably CsoS2 (Supplementary Fig. 6A, left); roughly 5% of purified CBs also contain a large unidentified protein complex (Supplementary Fig. 6A, right). Some CBs have fibrils

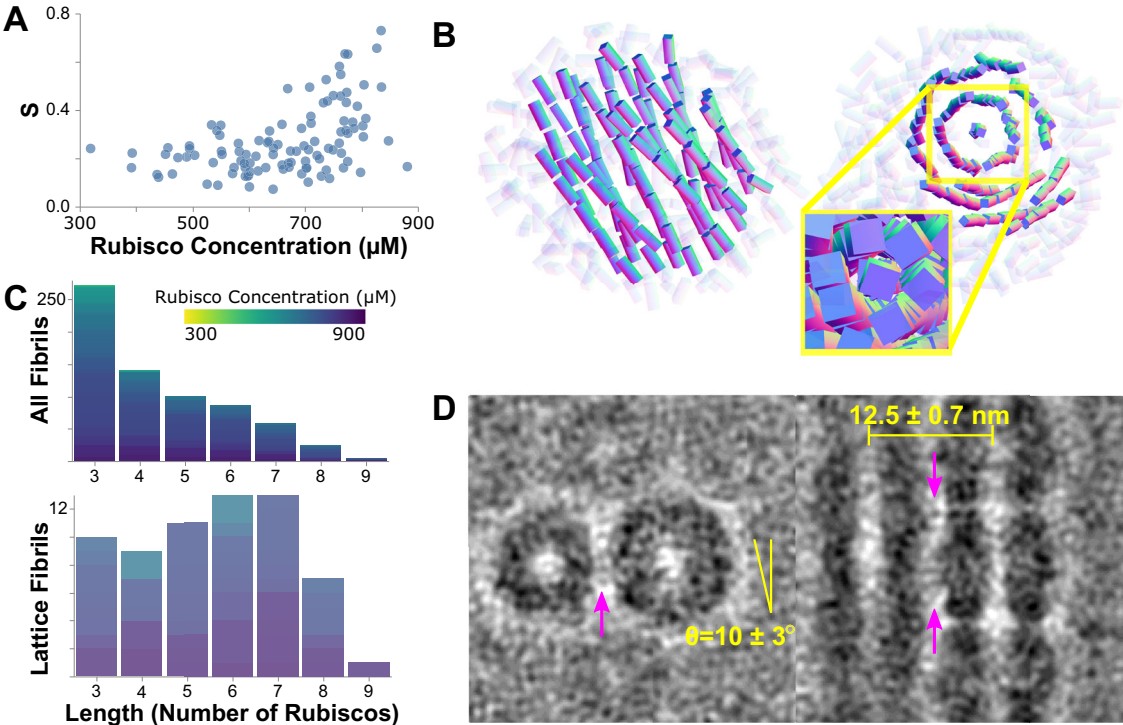

**Fig. 3 | Rubisco forms a twisted hexagonal lattice. A** The second rank-order scalar S indicates global alignment along one axis in ordered CBs at concentrations exceeding 650 μM. **B** A representative rendering of the ordered CB in Fig. 2A shows the Rubisco lattice from two angles. Rubisco complexes are rendered as rectangles with a slender side length for clarity; lattice Rubiscos are opaque and non-lattice Rubiscos are transparent. Inset: at-scale rendering showing the six-fold symmetry.

**C** Histograms of fibril length for all fibrils (upper) and inner-lattice fibrils (lower) show that lattice formation is associated with increased fibril length.
**D** Subtomogram averages show loose registration between the fibrils, with consistent spacing and tilt. Faint density in the fibril interface (magenta arrows) does not average to a specific interaction site.

arranged parallel to the shell instead of the single central lattice (Supplementary Fig. 6A, B, Fig. 2A). Finally, even in the most ordered CBs, a substantial portion of the Rubisco does not participate in the lattice (Fig. 3B). This heterogeneity is consistent with models of CB assembly, which are based on phase separation and binding affinities rather than a tightly ordered, regular assembly mechanism[13,15,26].

## Discussion

Previous in vivo and in vitro tomography studies observed apparent concentric shells of Rubisco inside α-carboxysomes, and used radial averaging to show that Rubisco formed a shell-adjacent layer[12,15]. However, the resolution in these studies was not high enough to determine Rubisco orientations necessary for ultrastructure analysis. Recently, tomographic analysis of the algal pyrenoid has shown that Rubisco behaves in a liquid-like fashion[10,27–29]. In contrast, our high-resolution analysis of organization in the CB indicates that Rubisco organizes into a lattice-like ultrastructure in some CBs.

We observed three Rubisco packing types in the CBs: sparse, dense and ordered (Fig. 2A). Sparse CBs lack the Rubisco concentration to form an ultrastructure, but the dense and ordered packings overlap in Rubisco concentration and both display Rubisco polymerization. The distinction between dense and ordered CB packing is whether the fibrils align, which supports fibril elongation and expansion of the nematic phase. It is likely that the CB interior can reorganize, given the wobbly Rubisco-Rubisco interaction (visible in the non-zero bending angle, which requires at least 2 of the 4 small subunit interaction sites to be unbound). In the β-CB system, characterization of reconstituted components and whole particles indicates that the interior becomes oxidized during β-CB maturation, with a concomitant increase in the mobility of interior cargo proteins[10,21,26]. There is biochemical evidence that the α-CB may mature through a similar

oxidation process[30,31], but the relationship between spatial heterogeneity, mobility and chemical environment remains unproven in this system.

The six-fold symmetry in the ordered Rubisco fibril packing contrasts with the four-fold symmetry of the Rubisco itself (Fig. 3B, Supplementary Movie 1). The symmetry break in situ between Rubisco's four-fold symmetry and its lattice may be facilitated by the twist along the fibrils, which combined with the large variance results in inconsistent lateral face presentation. This is in contrast to Form I Rubisco crystal structures in the PDB, which often have four-fold symmetric packing of parallel, untwisted fibrils (Supplementary Fig. 4). This four-fold crystal packing is likely selected against in vivo because it could interfere with Rubisco function by obstructing motion of loop 6 during the catalytic cycle[32], obscuring peptide binding sites, and limiting diffusion of substrates and products. In contrast, the six-fold fibril arrangement has at least 1 nm of space around the circumference of each enzyme. The six-fold lattice may therefore be a packing mechanism to preserve function at concentrations that could otherwise crystallize or sterically impede function.

An open question is the role of fibril formation in vivo. Polymerization is a highly effective packing strategy, and the six-fold lattice preserves function at concentrations that could otherwise sterically impede function. Rubisco polymerization may also assist in CsoS2-based Rubisco condensation during CB shell assembly, facilitating its incorporation at concentrations that can exceed some crystal structures and approach Kepler packing (the maximal density for packing of spherical objects). Importantly, the concentration dependence of the fibril formation is likely tuned to ensure polymerization only occurs when the local Rubisco concentration is increased by CsoS2 during shell encapsulation (Fig. 6A). Highly symmetric protein complexes have an innate tendency toward long fibril formation[33], and

uncontrolled cytoplasmic polymerization of Rubisco could disrupt CB formation and its essential metabolism[15]. Microcompartment bioengineering is a growing field, and this polymerization-based mechanism for efficient, functional, and controlled packing of enzymes may be useful for future discovery and designs in both CBs and other microcompartments.

While this paper was in revision, a preprint was released containing tomography and subtomogram averaging of purified CBs[34,35]. These results are broadly consistent with ours, including the lack of a physical tether between Rubisco fibrils. That study assigned density to CsoS2 in Rubisco complexes located near the shell, which may result from different classification or thresholding compared to our analysis for this lower-occupancy interaction[17]. CBs were also treated with super-physiological calcium concentrations which resulted in the formation of an alternate lattice with different symmetry[34], consistent with our prediction that the physiological lattice we have characterized is dynamic and capable of rearrangement.

The higher-order arrangement of enzymes is often a component of their function and regulation[36-38]. Rubisco itself has a rich representation in complex ultrastructures, from plant chloroplasts to algal pyrenoids to bacterial carboxysomes. The Rubisco ultrastructure within the *H. neapolitanus* CB is novel in its design of loosely structured polymers rather than either a liquid-like encapsulation or a highly regular ultrastructure. Rubisco thus provides a unique comparative system for investigating the physical organization of the cell. Future studies will be needed to further elucidate the mechanisms underlying the observed divergent ultrastructural arrangements and their role in the control and regulation of this important enzyme.

## Methods

### Carboxysome preparation

We purified α-carboxysomes (CBs) from wild-type *H. neapolitanus*[6]. Briefly, wild-type *H. neapolitanus* cells were grown in DSMZ-68 medium in a 10 L chemostat (pH 6.4, ambient air sparged, 30 °C), harvesting cells every 2–3 days by 6000 × *g* centrifugation. A ~8 g cell pellet (resulting from 10–15 L of culture) was chemically lysed and purified by centrifugations (12k × *g* 15 min, 40k × *g* 30 min) and sucrose gradient (25 mL, 10–50% sucrose step gradient, centrifuged 105k × *g*

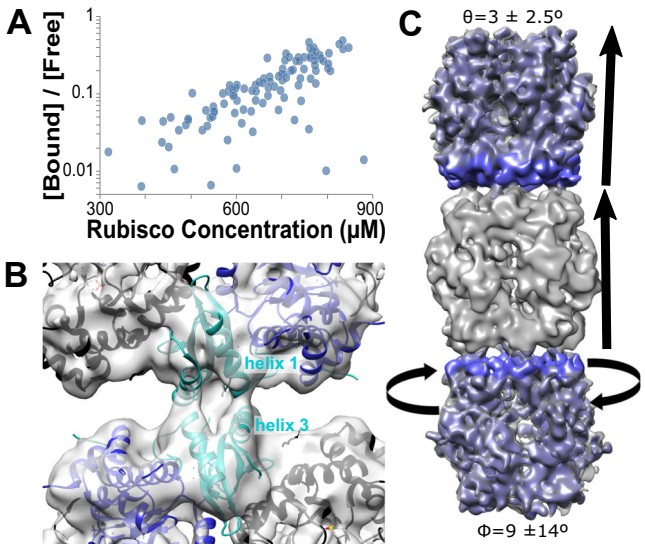

**Fig. 4 | Rubisco polymerizes to form fibrils. A** Rubisco-Rubisco binding scales with concentration. **B** A 10 Å slab through the subtomogram average of the Rubisco-Rubisco interface with the docked 1SVD crystal structure shows an interaction between helices 1 and 3 of the small subunit (cyan; large subunit, blue; additional subunits, black). **C** A subtomogram average of Rubisco fibrils (gray) shows variable tilt (θ) and twist (φ) in the interaction. A difference map of the fibril and free Rubiscos shows density only in the Rubiscos above and below (blue), with no other protein mediating the interaction.

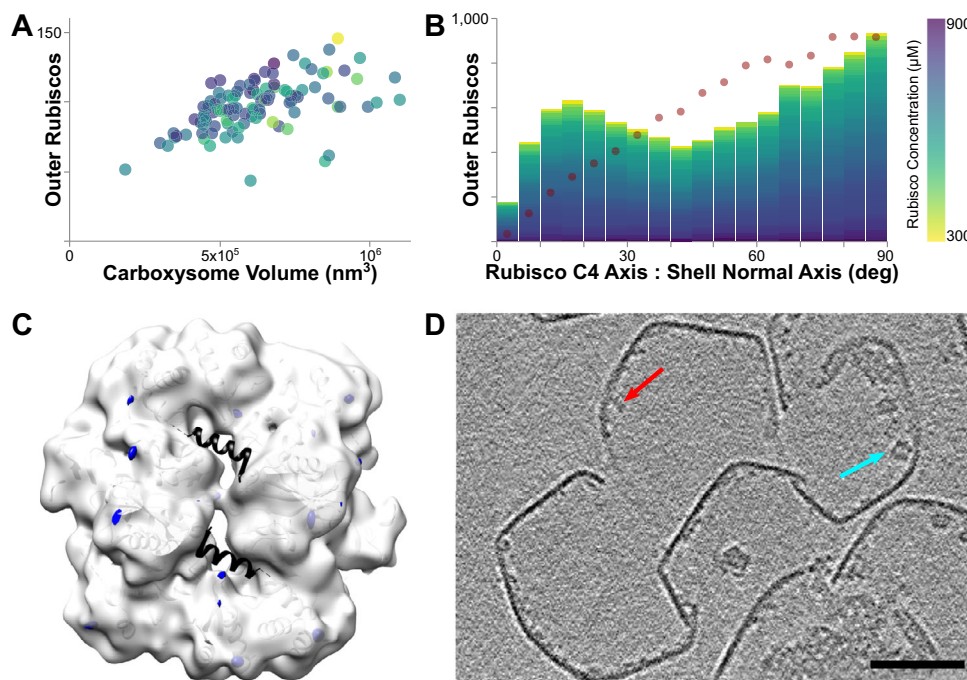

**Fig. 5 | Rubisco in the outer layer adjacent to the shell. A** Rubisco lines the shell in a volume-dependent but concentration-independent manner. **B** A histogram of the angle between the Rubisco C4 axis and the normal axis of the adjacent shell shows a modest preference for perpendicular orientations. Dotted line indicates random orientation. **C** We do not observe strong CsoS2 occupancy in the outer Rubisco layer. White: C1 interior Rubisco subtomogram average (3 nm slab, filtered). Blue: difference map showing density added in the outer layer Rubisco subtomogram average. Black: 6UEW crystal structure with CsoS2 peptide. **D** A 4.4 nm-thick orthoslice of shell-associated density in broken shells. Red arrow, example unassigned protein density; cyan arrow, Rubisco. Scale bar 50 nm.

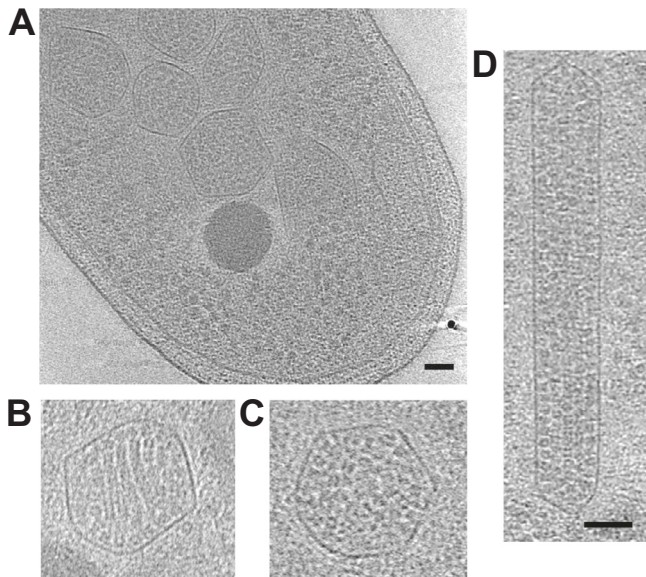

**Fig. 6 | Carboxysomes inside *H. neapolitanus* cells. A** An orthoslice through a tomogram of *H. neapolitanus* shows multiple carboxysomes packed inside the center of the cell. CBs are frequently in close proximity to phosphate granules and show greater heterogeneity compared to purified samples. **B** An ordered CB in vivo. **C** A dense CB in vivo. **D** Some CBs display an elongated morphology, with Rubisco aligned in rows with or without polymerization. All scale bars 50 nm; all orthoslices 5.4 nm thick.

35 min). Quality and purity of sample was analyzed by Coomassie stained SDS-PAGE and negative stain TEM. The final sample had an A280 of roughly 10, and was stored at 4 °C in TEMB buffer (10 mM Tris pH 8, 1 mM EDTA, 10 mM MgCl₂, 20 mM NaHCO₃) until use.

### Tomography

10 nm gold colloids were mixed with 5% BSA in PBS to passivate the surface, then were buffer-exchanged into TEMB buffer and concentrated. The CB sample was diluted with the BSA-coated gold and plunge-frozen onto glow-discharged C-flat 2/2-300 EM grids using a Vitrobot. Imaging was performed on a Titan Krios with a Gatan K3 camera and energy filter running Digital Micrograph. Tomograms were collected in SerialEM according to Supplementary Table 2. Frame alignment, 2D image processing, and tilt alignment were performed in etomo; CTF estimation in ctffind4; and tomogram reconstruction used novaCTF with the phaseflip option[39–41]. Dose weighting was performed after defocus estimation but before tilt alignment[42].

For cellular tomograms, *H. neapolitanus* cells were grown in 20 mL DSMZ media, pelleted by mild centrifugation, and resuspended in 0.5 mL DSMZ media. Cells were treated with 0.25 mg/mL ampicillin for 60 min at room temperature[43]. Immediately following antibiotic treatment, cells were mixed with gold colloids and plunge-frozen in liquid ethane as above. Tomograms were collected according to Supplementary Table 2. Summed-frame tilt stacks were reconstructed in etomo and the CBs classified manually.

### Subtomogram averaging

Particles were identified in SIRT-like filtered tomograms using a customized workflow designed to identify every Rubisco complex within each CB. The center of the CB shell was manually segmented in the xz and xy planes in IMOD[44], and a three-dimensional grid of Dynamo model points spaced 30 bin2 pixels apart was generated to fill this space using the Matlab inpolygon function. One iteration of subtomogram alignment was performed in Dynamo in bin4 against a reference average of Rubisco complexes outside the CB, with

translation searches limited[45]. Duplicate positions were removed, then the revised model positions were sent to Relion for extraction and classification[46]. Out of four classes, Relion consistently grouped subtomograms as Rubisco, shell or noise. This approach yielded Rubisco particles with an estimated 15–20% false positive and false negative rates. Classified particles were manually screened using a rule-based approach, eliminating Rubisco identifications outside the shell, shell identifications on Rubisco fibrils, identifications too close together, and so on. Screening was performed conservatively.

Following manual screening, the estimated false negative and false positive rates fell to 1–2% of total Rubisco particles within a CB when compared with manual identification in a clear, high-defocus tomogram. Unfiltered map resolutions were not degraded by using every single particle identified (Supplementary Fig. 7), further strengthening the evidence for the strong success rates in particle identification. Resolution was calculated using a loose mask in Dynamo; final mask-corrected FSC was calculated in Relion using a two-pixel extended mask derived from the low-pass filtered 1SVD PDB structure (https://www.rcsb.org/structure/1SVD, Supplementary Fig. 7). For calculating resolution according to number of particles included, particles were first sorted by cross correlation to the final half-map, and the best-correlating particles were used according to the target percentage.

The general subtomogram averaging workflow was based upon a previous high-resolution approach from the Briggs group[47]. Following particle identification, subtomograms were re-extracted from novaCTF weighted back projection tomograms. CBs were divided into half-sets, and subtomogram alignment and averaging was performed by progressively moving a lowpass filter to the resolution at which the FSC crossed 0.143 in the previous iteration[45]. The binning and search space were decreased as resolution improved, and C4 symmetry was imposed after searches were restricted to <90-degree rotations. The best 80% of particles according to cross-correlation were used to generate the reference for the next iteration, but all particles were aligned for use in ultrastructure analysis. Masks throughout were kept loose, with no added space for CsoS2 or CsoSCA peptides. When the resolution ceased to improve by at least 1 Fourier pixel per iteration, the half-maps were sent to Relion for post-processing and final resolution calculations[46]. Rigid body docking was performed in UCSF Chimera[48].

Subtomogram averages of fibrils, monomer Rubisco, and outer and inner layer Rubisco were calculated in Dynamo using the orientations and positions from the full-dataset alignment, C1 symmetry and no masking (see Data Analysis for classification approach). The outer and inner layer analyses were processed in half-sets with standard FSC calculations, but for visual clarity Fig. 5C uses low-pass filtered averages of both half-sets. The fibril average was computed from all particles and low-pass filtered to 1 nm due to particle number constraints. Difference maps were calculated by subtracting the average of unbound, non-outer layer Rubisco from the average of choice.

### Data analysis

Forty-one tomograms of intact *H. neapolitanus* cells and 62 tomograms of purified CBs were collected. CBs were manually classified as sparse (>30% empty volume), dense (<30% empty volume) or ordered (containing a bundle of at least 6 fibrils).

Cellular tomograms were used for visualization and manual morphological classification only. The sixty-two tomograms of purified CBs contained 32,930 identified Rubisco complexes within 139 CBs. All of these data were used for subtomogram averaging. Further data analysis was performed on CBs wholly within the field of view in all tilts, and with defoci allowing clear visual contrast (typically −2 μm or more), for a subset of 26,224 particles in 107 CBs.

CB volume calculations utilize a previous observation that Rubisco forms an outer layer immediately adjacent to the shell[12,15]. This

behavior allows treatment of the Rubisco as a sphere in an accessible volume calculation. We defined the outer layer of Rubisco as the positions used in a convex hull calculation, and calculated the approximate volume contained by calculating the volume enclosed by each three-Rubisco facet plus a fourth reference point (a non-hull Rubisco position). Total Rubisco concentration was then calculated by converting Rubisco complexes per $m^3$ to µM.

All data analysis calculations consider the D4 symmetry of the Rubisco complex. Dynamo uses a ZXZ' Euler angle convention. Subtomogram averaging placed the C4 axis of the complex along the Z axis in Dynamo, simplifying the handling of symmetry in the analysis. Only the Rubisco-Rubisco twist calculation considers the final azimuthal rotation (Z'); all other calculations only consider the orientation of the Rubisco C4 axis, using the orientation vector defined by the ZX Euler angles. Angles between Rubisco complexes are calculated using the formula

$$\theta = \arccos \left| \frac{\mathbf{v}_i \cdot \mathbf{v}_j}{\|\mathbf{v_i}\|\|\mathbf{v_j}\|} \right| \tag{1}$$

where $v_i$ and $v_j$ are the vectors defined by the C4 axis of two Rubisco complexes, and the absolute value of the dot product accounts for the identical head and tail due to the D4 symmetry. For calculations comparing the C4 Rubisco axis and the shell normal vector, the shell normal vector was calculated using a plane fit to the nearest facets in the convex hull calculation.

To search for Rubisco fibrils, we looped through every Rubisco centroid and set a search point 45 pixels along the C4 axis in both directions. Rubisco complexes are considered bound to the reference Rubisco if they have centers within 22.5 pixels of the search point and a C4 axis orientation differing <25 degrees from the reference. A Rubisco fibril is considered as inside a lattice if it is surrounded by 6 other fibrils within 70 bin2 pixels. A Rubisco is considered as being in the outer layer if it participates in the convex hull calculation.

Because the outer layer of Rubisco has location-specific properties, outer layer Rubisco complexes are not counted in fibril length even if they appear to participate. Outer layer Rubisco complexes are also excluded from the nearest-neighbor alignment analysis (Supplementary Fig. 3). The nearest-neighbor analysis does not use total Rubisco concentration, but rather scales relative concentration of non-outer layer Rubisco (calculated as total inner Rubisco complexes divided by the CB volume).

The lattice visualization was performed in THREE.js. Rubisco alone or in a two-member fibril was made transparent, and the diameter was decreased to visually emphasize the pattern.

The collective alignment of Rubisco in the CB was estimated by calculating the second order tensor, $Q_{ij}$, where v is the orientation vector of a Rubisco complex and $\delta$ is the Kronecker delta[23].

$$Q_{ij} = \left\langle \mathbf{v}_i \mathbf{v}_j - \frac{1}{3} \delta_{ij} \right\rangle \tag{2}$$

The scalar, $S_{ij}$, is calculated from the largest eigenvalue of the tensor. It has a value range of 0 to 1, and the boundary between isomorphic and nematic phases occurs around 0.3–0.4.

$$S_{ij} = \frac{3}{2} \lambda_{max}(Q_{ij}) \tag{3}$$

For means and errors presented in the text, all histograms were visualized and all CBs are included in the analysis unless otherwise noted. Distributions appearing to be Gaussian and with no scientific reason to assume otherwise are presented as a Gaussian mean and standard deviation. The bend angle of the Rubisco fibril, which shows a long-tailed distribution, is presented as the mode and full width half maximum. Plots showing histograms of angle orientations also display random axis orientations, plotted with the same histogram binning and symmetry consideration[49].

We evaluated the efficiency of Rubisco packing within CBs by comparing a calculated maximal packing with actual packing density. The twisted hexagonal fibril arrangement cannot tile 3D space indefinitely, precluding definition of a unit cell. To estimate the lattice density we centered on rubiscos in the middle of fibrils containing six or more rubiscos and with six neighboring fibrils and then sampled the mass density in the vicinity (up to 23 nm away). The rubisco coordinates from PDB 1SVD have unmodeled residues at the termini and in some loop regions. To obtain a full atomic model including hydrogens we used Alphafold2 to generate a model (rmsd 0.4 Å). This model was fit to the subtomogram template density using the phenix.dock_in_map utility and then transformed into the search space using the Dynamo coordinates and orientations in tandem with rubisco's symmetry operators. The mass density was calculated as the median density in the subvolumes surrounding the central rubisco coordinates. The density was also calculated using a coarse-grain procedure in which the rubiscos were replaced with 6 nm radius spheres of uniform density. The results of this had close correspondence to the atomic model and also enabled a comparison to Kepler packing, a maximally dense hexagonal close pack.

## Conservation analysis

Candidate CB loci were selected from the Integrated Microbial Genomes database on the basis of having a co-occurrence csos2 (PFAM: PF12288), rubisco large and small subunits (PF00016 and PF00101), α-carboxysomal carbonic anhydrase (PF08936) and bacterial microcompartment shell proteins (PF00936) within 100 kb. Among these sequences the likely α-carboxysomal rubisco genes were conservatively selected as those with consecutive small and large subunits within 12 ORFs of csos2. To reduce bias, the sequences were clustered at 95% sequence identity of CbbL using uclust. After this filtering process the CbbS and CbbL proteins (N = 47 pairs) were aligned at the amino acid level with mafft and analyzed with Weblogo 3.7.10.

## Sample preparation effects

All high-resolution analyses were performed on purified CBs to maximize resolution and throughput. Purification selects for pseudo-icosahedral CBs in a band of diameters and imposes mechanical stresses[15]. While CBs remain functional following purification, damage from this procedure could artificially reduce Rubisco concentrations (the sparse CB set) or distort native ultrastructure (increased standard deviations in parameters such as fibril tilt and bend). Sparse CBs, though possibly a purification artifact or over-sampling, provide important low-concentration datapoints for Rubisco binding interactions and confirmation of the likely tether between the shell and the outer layer of Rubisco. For this reason, sparse CBs were included in the dataset despite their rarity in vivo. Lattice analyses focus on the ordered CBs, which are qualitatively similar to the ordered CBs inside cells. All analyses are performed on all CBs, and only limited by local Rubisco interactions (minimum fibril length, maximum lateral distance to next fibril, etc.).

The in vivo data are subject to greater noise in the visual classification, and more limited angular space for clear viewing. The bias in the in vivo dataset favors dense CBs, which may be over-estimated compared to ordered CBs due to difficulty in clearly visualizing packing in the cellular interior at small angles relative to the electron beam. Tomogram areas too thick to clearly assess Rubisco orientations were not analyzed.

## Statistics and reproducibility

62 tomograms of purified CBs and 41 tomograms of *H. neapolitanus* cells were collected, each on a single grid during one data collection

session. Two independent biological preparations of purified CBs and four independent preparations of cells were prepared during screening of sample preparation conditions, and appeared qualitatively consistent. Morphological classifications were performed on the entire dataset (Supplementary Fig. 1, Supplementary Table 1).

Analyses of purified CBs examine 26,224 Rubisco particles in 107 CBs. This dataset contains 5011 Rubiscos participating in 690 fibrils of three or more subunits, and 10,365 shell-adjacent Rubiscos. Parameters are reported as mean and standard deviation unless otherwise indicated. Standard deviation is only used if a histogram shows a Gaussian-appearing distribution.

### Reporting summary
Further information on research design is available in the Nature Research Reporting Summary linked to this article.

## Data availability
The data that support this study are available from the corresponding authors upon reasonable request. The Rubisco unfiltered half-maps and full filtered map generated in this study have been deposited in the Electron Microscopy Data Bank under accession code EMD-27654. The complete set of subtomograms with position information as well as sample tomograms and frames generated in this study have been deposited in the Electron Microscopy Public Image Archive under accession code EMPIAR-11125. Raw tomogram movie frames generated in this study have been deposited in the Caltech Electron Tomography Database (https://etdb.caltech.edu/).

## Code availability
All algorithms used in tomogram reconstruction and subtomogram averaging are available through the referenced software packages. Subtomogram analysis code is available at https://observablehq.com/collection/@lametskas/cbpaper. Conservation analysis is at https://github.com/trogon44/carboxysome_tomography/tree/master/rubisco_conservation, and packing comparisons between ideal lattices, crystal structures and CBs is at https://github.com/trogon44/carboxysome_tomography/tree/master/rubisco_density[50]. All code is supplied as-is.

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

## Acknowledgements

Cryo-electron microscopy was done in the Beckman Institute Resource Center for Transmission Electron Microscopy at Caltech, and sub-tomogram alignment and averaging used the Caltech Resnick High Performance Computing Center. We thank S. Chen and A. Malyutin for assistance with tomography data collection, A. Burt for pseudocode to transition between Dynamo and Relion software packages, and A. Pranger for assistance reconstructing cellular tomograms. This work was supported by a Ruth L. Kirschstein NRSA Individual Postdoctoral Fellowship F32 1F32GM135994-01 to L.A.M., NIH R01GM129241 to D.F.S., and NIH R01 AI127401 to G.J.J.

## Author contributions

C.B., T.G.L. and L.M.O. purified the carboxysomes and provided *H. neapolitanus* cells. L.A.M. performed cryo-electron tomography and subtomogram averaging; L.A.M., D.O. and D.R.L. analyzed the tomography data. L.M.O. performed conservation and crystal packing analyses. L.A.M., L.M.O., C.B., D.F.S., and G.J.J. designed research, interpreted results, and wrote the manuscript.

## Competing interests

The authors declare no competing interests.
