## [Peer Review File · Nature Communications]

Rubisco forms a lattice inside alpha-carboxysomesReviewers' Comments:

Reviewer #1:

Remarks to the Author:

I very much enjoyed reading this manuscript, which describes an interesting set of data relating to packing arrangements of Rubisco in *Halothiobacillus neapolitanus* carboxysomes. It utilizes cryo electron tomography, coupled with existing Rubisco crystal structures, to solve Rubisco arrangements within purified carboxysomes. A take-home result is that high concentrations of Rubisco within carboxysomes leads to increasingly ordered packing of the enzyme, via observable interaction interfaces. While the manuscript provides very interesting insights into Rubisco ordering in these microcompartments, I felt that more could be done to extend the impact of the manuscript and further elucidate the observed interactions between Rubisco holoenzymes. For example, can the interactions be eliminated through mutation of the interacting interfaces? Are the interacting interfaces highly conserved? What would be expected in carboxysomes from different organisms given the conservation (or lack thereof)?

A general question relating to the observed 'classes' of carboxysomes based on their relative packing density (sparse, dense, ordered): could the observed heterogeneity in cbx composition and structure be due, in part, to the growth conditions of Halo and/or the potential lack of growth dependency on cbx rubisco in these organisms (they can grow heterotrophically and have more than one Rubisco). Would a system or a growth condition which enforces cbx growth dependency provide more definitive results? Also, is there a way of confirming that the observed interactions are not artefactual? It is worth noting that similar tightly packed Rubisco, appearing as a paracrystalline array, have been observed in beta carboxysomes (e.g. Kaneko et al.) and in older EM images of pyrenoids, both have which have since been described as liquid condensates which can become even more freely moving within oxidised beta carboxysomes, suggesting that these previous paracrystalline observations were potentially artefactual. Can the authors provide comment to assure the reader the structures observed here are a genuine reflection of functional alpha-cbx?

In general, I think the work is highly interesting but would be suitable for this journal if additional analyses were carried out, as highlighted in specific comments below. I would be extremely happy to review a version of the manuscript with amendments.

Specific comments:

Line 36: An additional reference for consideration here is the recent review by Hennacy and Jonikas (Annual Review of Plant Biology), further highlighting the importance of transplanting such systems to a non-native host.

Line 40-41: the alpha-cbx of Halo is a 'representative' of cbx's found also in cyano's and other chemoautotrophs?

Line 44: 'could potentially', is the evidence for the Rubisco-S2 interaction inconclusive?

Fig 1. Gene names should be italicized but not capitalized. Alternatively, protein names should be capitalized and not italicized.

Line 56: 'non-equilibrium bicarbonate pool'. This is not clear for those who do not understand CO₂ chemistry. Perhaps expand to indicate that there is specifically a disequilibrium between CO₂ and bicarbonate, in favour of bicarbonate, within the cytoplasm?

Line 58-61: Models including Kaplan, Badger, Mangan, Long etc. seem to suggest that encapsulation is sufficient. However, all these models are based on unknown diffusional limitations of the shell. Therefore, the primary unknown is the real diffusional resistance of the shell and does not appear to necessitate allosteric regulation of kinetics.

Line 66-67: What evidence is presented to suggest this unique feature leads to the rate of CO₂ fixation? Comparison of active site turnover numbers for the different packing classes of carboxysomes would provide this information. Purification of numerous carboxysome preparations from cells grown under a variety of conditions may lead to different proportions of packing types, therefore enabling such analysis. Correlation of packing type and site-dependent Rubisco turnover rates would clarify if packing arrangement led to differential (and possibly allosteric) regulation of Rubisco function.

Fig 3. A 'fibril' needs defining more clearly. I also don't fully appreciate sub-fig B. Could it be superimposed on a tomogram image for context?

Fig 4. Is the observed interaction between Rubisco's (Fig 4B) consistent with observations relating to any known Rubisco mutations, or have such mutations been assessed? Attempts to address this would make for more impactful results, as would some assessment of whether the observed interaction affects Rubisco catalysis.

Fig 5. This observation is highly relevant to observations of rigid rod-like carboxysomes from Halo (Cai et al., *csoS4* deletions) and in the chloroplast expressed carboxysomes of Long et al. where it is possible that only shell-bound Rubisco's are present, and they appear to have a specific arrangement. How do the observations here compare? Extended Fig 4 shows the 16 Angstrom offset which may correlate very well with observed packing in rods, possibly contributing to rod-formation by an iterative extension of shell formation and Rubisco organization. Again, this highlights an opportunity to assess this via mutational analysis to manipulate the Rubisco interacting interfaces. In addition, how conserved are the proposed interacting interface residues? It would be safe to assume that there should be a high degree of conservation in these regions of Rubisco large and small subunits.

Fig 5D. Please identify what is assumed to be CsoSCA and Rubisco in this image, as pointed out in line 195-198.

Lines 122-129: This short paragraph seems to take some time to let the reader know that S2 is possibly the link to Rubisco here. Perhaps it could be written more succinctly and definitively.

Lines 174-180: Is there an observed Rubisco:CsoS1 interface? Would observation of rod-like cbx's offer any clues?

Line 202-203: a large 'enzyme' consistent in size and shape with a 20S proteasome? No evidence is presented for what appears to be a disordered aggregate of protein, I assume in the first image in Ext. Fig 5.

Line 223-225: Note that b-cbx have been observed as paracrystalline in cyanobacteria (Kaneko et al, via TEM), despite current knowledge that the interior of these cbx's should be more motile. This raises the question of whether the observations here and in many circumstances might result as artefacts from the sample preparation process or if they are indeed real. This is where further analysis could be extremely helpful. For example, can specific Rubisco mutations still enable entrapment of the enzyme in carboxysomes but eliminate organised structure at high internal concentrations?

Line 246-247 is a key outcome here. It implies that an evolved, preferential high-concentration-dependent packing will still maintain Rubisco function under 'saturating' carboxysome filling conditions. This should be emphasised, and it implies a calculable upper limit of Rubisco concentrations within carboxysomes which is driven entirely by the observed arrangement.

How does this observed Rubisco packing compare with what has been described as 'Kepler packing' as an estimate of maximum Rubisco quantities in both alpha and beta carboxysomes?

Extended Fig 1. Is it possible to delineate, on these data plots, the apparent 'classes' of cbx (dense, sparse, ordered)?

Reviewer #2:

Remarks to the Author:

"Rubisco forms a lattice inside alpha-carboxysomes"

Reviewed by Manon Demulder and Benjamin Engel

Metskas et al. use cryo-electron tomography (cryo-ET) to provide the first high-resolution visualization of the alpha-carboxysome, which has remained rather enigmatic compared to the better understood beta-carboxysome. Surprisingly, the authors observe linear fibrils of Rubisco within some of the isolated carboxysomes. They then generate a subtomogram average of this Rubisco at a fairly impressive 4.5 Å resolution to show that fibril formation is apparently mediated by interactions between the Rubisco small subunits. The polymerization of Rubisco into fibrils is a novel and interesting result, which is of high interest to the whole field of carbon concentration/fixation. The

transition from Rubisco liquid condensate into ordered fibril is also of broad interest to the field of phase separation, and may even carry potential interest all the way out to fields such as neurodegeneration, which study these liquid-to-solid state transitions. However, there is one major issue with this study, which should either be addressed with additional experiments, or be acknowledged with careful rewording of the text and its interpretations.

Major issue:

A major limitation of this study is that the cryo-ET was performed on isolated carboxysomes. Some features such as the Rubisco fibrils are likely also present within the cell, as I cannot imagine how isolation of carboxysomes would induce fibril formation. However, the isolation procedure and subsequent blotting onto EM grids are unquestioningly disruptive-- this is clearly shown by the aggregate of broken carboxysome shells in Fig. 5D. Furthermore, I know from my own experience that larger isolated structures (e.g., centrioles) are very easily compressed when they are blotted and frozen into a thin layer of ice on an EM grid. This compression is unavoidable, and is often dramatic. Compression would almost certainly impact the organization of Rubisco within the isolated carboxysomes, especially given the flexibility/disorder of the CsoS2 linker. It could also detach Rubisco in the carboxysome interior from the layer of Rubisco more tightly bound to the carboxysome wall. The authors conclude that "the most consistent feature of the CBs is their compositional and structural heterogeneity" (Line 200), but with the present data, it is not possible to say whether this is a result of carboxysome biology or an artifact of the sample preparation. Specifically, the isolation and blotting of carboxysomes could potentially affect the following analysis presented in this study:

1) The relative abundance of Rubisco fibrils (Figs. 2A, 4A)

Line 94: "Roughly one third of the carboxysomes displayed ordered packing"; Line 131: "The ratio of bound and free Rubisco concentrations scales with Rubisco concentration in the CB (Figure 4A)."
-> Are these numbers representative of the biology or the sample preparation?

2) The bending angle and twist between Rubisco complexes along a fibril (Fig. 4C)

Line 140: "There is a bending angle of 3 ± 2.5 degrees and a 14-degree standard deviation in the twist between stacked Rubiscos."
-> Is this fibril bending and twisting present in the cell, or is it an artifact of compression forces caused by the blotting?

3) The higher-order packing of fibrils into a twisted hexagonal lattice (Fig. 3)

Line 117: "Rendering all Rubisco fibrils within a CB displays the ordered phase: a loose hexagonal lattice-like ultrastructure of Rubisco fibrils twisting about each other with six-fold pseudo-symmetry (Figure 3B). The fibrils are held at a distance of 12.5 ± 0.7 nm with a tilt of 10 ± 3 degrees (Figure 3D)."
-> Again, is this lattice spacing and tilt affected by the blotting forces?

4) The packing and orientations of non-fibril Rubisco in carboxysomes (Extended Figs. 1, 3)

Line 95: "Other CBs displayed a range of Rubisco concentrations, from sparse to dense (Extended Data Figure 1). A nearest-neighbor Rubisco alignment search across all CBs revealed a concentration dependence: Rubisco orientation is random at low concentration (sparse CBs), but becomes increasingly non-random as concentration rises (dense CBs, Extended Data Figure 3)."; Line 205: "Finally, even in the most ordered alpha-carboxysomes, a substantial portion of the Rubisco does not participate in the lattice (Figure 3B). This heterogeneity is consistent with models of CB assembly, which are based on phase separation and binding affinities rather than a tightly ordered, regular assembly mechanism"

-> Is the observed heterogeneity in Rubisco concentration and organization representative of biological variability or artifacts from the isolation and blotting?

5) Rubisco organization and concentration next to the carboxysome shell compared to the carboxysome interior (Fig. 5)

Line 174: "The Rubiscos adjacent to the shell occasionally participate in the long fibrils, but not consistently, and appear to behave differently than Rubisco in the CB interior."; Line 176: "the layer is well-populated even in otherwise sparse CBs. We analyzed the angle of the Rubisco C4 axis relative to the shell and found a predominantly random distribution... (Figure 5B)."

-> Is this distinction in the organization of shell-adjacent Rubisco also seen within native cells, or have forces from the isolation and blotting detached Rubisco fibrils in the carboxysome interior from the layer of Rubisco more tightly bound to the carboxysome wall?

6) Proposals about carboxysome assembly/maturation mechanisms (Discussion)

Line 219: "Sparse carboxysomes lack the concentration necessary to form an ultrastructure, but the dense and ordered packings overlap in Rubisco concentration. All packing types display polymerization of the Rubisco; therefore, the primary distinction between dense and ordered CB packing is whether the fibrils align"; Line 226: "Therefore, the phase transition between dense and ordered packing may involve the CB maturation trajectory."

-> Are we really seeing a maturation-driven phase transition, or are some carboxysomes more disrupted than others?

I am surprised and a bit disappointed that the authors did not complement this analysis of isolated carboxysomes with some bona fide in situ data of FIB-milled *Halothiobacillus* cells. This approach is becoming mainstream for leading cryo-ET labs, and indeed, the last author of this study has access to the necessary FIB/SEM instrumentation, as evidenced by their recent publications:

Swulius et al. 2018 <https://doi.org/10.1073/pnas.1711218115> ;
Martynowycz et al. 2019 <https://doi.org/10.1016/j.str.2018.12.003> ;
Zhang et al. 2020 <https://doi.org/10.1126/sciadv.abc8258> ;
Carter et al. 2020 <https://doi.org/10.1126/sciadv.aay9572> ;
Mageswaran et al. 2021 <https://doi.org/10.1101/2021.12.13.472487> ;
Nicolas et al. 2022 <https://doi.org/10.1101/2022.01.31.478342> ;

Regardless of issues associated with the isolation procedure, the Rubisco fibrils appear to be real, and they are a novel finding of high interest to the field. As far as I can tell, the subtomogram averaging performed on these fibrils is methodologically solid, providing some clues into how the fibrils may form via interactions between the small subunits. Thus, I see two possible routes to publication: 1) The authors explicitly describe this work as an in vitro study of isolated carboxysomes. They emphasize the aspects that likely remain true in these isolated conditions, such as the Rubisco-Rubisco structural interactions that establish the fibrils, while the other analyses that may be affected by the isolation (outlined above) should be carefully qualified. 2) The authors add complementary in situ cryo-ET from FIB-milled cells. This does not have to be an extensive dataset, but in situ analysis would be highly valuable to confirm the in vitro observations and add reliable information about the prevalence of different Rubisco organizations within native cells. I think route #2 is definitely preferred and will significantly increase the impact of this study. However, if the authors opt for route #1, I would not object to publication as long as the conclusions are properly qualified.

Minor issues and requested text changes:

Line 22, Abstract: "Here, we use cryo electron tomography and subtomogram averaging to determine an in situ structure of Rubisco at 4.5 Å"

-> This is certainly NOT an in situ structure. In situ means within the original natural environment. Specifically regarding cryo-ET, in situ means inside the cell, as has been demonstrated in many previous cryo-ET studies of intact small cells or larger cells that have been thinned with a cryo-microtome or by focused ion beam milling. The Rubisco subtomogram average in this study is derived from isolated carboxysomes, which are not in their native state.

Line 29, Abstract: In the abstract, it is claimed that the insights into Rubisco fibril formation "provide future directions for bioengineering of microcompartments."

-> In the discussion section, the authors should provide some detail on how Rubisco fibrils open these new bioengineering opportunities. Conversely, if it is too early to predict these possible applications, this claim could (should!) be removed from the abstract.

Line 66, Introduction: "This unique feature of the alpha-carboxysome likely helps to increase the rate of carbon fixation inside the CB."

-> How do the authors know that Rubisco polymerization is a unique feature of alpha-carboxysomes? Beta-carboxysomes also contain a pseudo-crystalline lattice, which has not yet been studied at high resolution. Also, the authors observe different degrees of order with similar Rubisco concentrations, so fibrils aren't simply a mechanism for increasing the concentration of Rubisco within the carboxysome (Line 219: "the dense and ordered packings overlap in Rubisco concentration"). Thus, without mechanistic studies specifically disrupting the fibrils, it is probably too early to conclude that Rubisco polymerization likely helps increase the rate of carbon fixation.

Line 117, Results: "Rendering all Rubisco fibrils within a CB displays the ordered phase: a loose hexagonal lattice-like ultrastructure of Rubisco fibrils twisting about each other with six-fold pseudo-symmetry (Figure 3B)."

-> How often was this hexagonal lattice observed in the dataset of 139 carboxysomes? Figs. 3A and 3C appear to analyze many carboxysomes, but the prevalence of this lattice is not clear to me. The figure legend claims that Fig. 3B is a representative example. In the supplement, please show at least four more examples of such hexagonal lattices in other carboxysomes.

Line 119, Results: "The fibrils are held at a distance of 12.5 ± 0.7 nm with a tilt of 10 ± 3 degrees (Figure 3D)."

-> Is the packing and helical tilt between neighbor fibrils always the same, or are there variations between carboxysomes? If not done already, please quantify this for the full dataset (It's not clear to me whether these numbers are averaged from every hexagonal lattice in the dataset or just from the example in Fig. 3B).

Line 124, Results: " We hypothesize that a disordered, lower-occupancy binding partner may maintain a maximum distance and promote tilt between fibrils. Such a linker would likely not be visible in a subtomogram average due to the combination of low occupancy and disorder, especially if there were also disorder in the binding site, causing the system to act as a 'fuzzy complex'. CsoS2 has the appropriate disorder, length and Rubisco binding sites to serve this role."

-> Wouldn't a low-occupancy disordered linker promote significant flexibility between the Rubisco fibrils instead of this specific spacing and tilt? Perhaps there is a more physical packing explanation for the parameters of the observed hexagonal lattice?

Line 165, Results: "Rubisco crystal structures also display variable interactions in this region. H. neapolitanus Rubisco structures 1SVD and 6UEW both show alternative longitudinal interactions in the crystal structure involving helix 3 (Extended Data Figure 4). Both crystal structure interactions lie within the range represented in our data, suggesting these conformations may be sampled in vivo (though not the dominant conformation)."

-> Two comments here. First, please be careful with the wording because isolated carboxysomes are not in vivo. Second, it would be helpful to provide some context for these two PDB structures. 1SVD is an apo structure, whereas 6UEW is bound by CsoS2 N-peptide, but this is not clear from the text.

Line 187, Results: "7 Å C1 interior Rubisco subtomogram average" (shown in Fig. 5C)

-> This Rubisco average appears to be filtered to a resolution that is too high, resulting in noisy spikey extensions all over the structure. Likely the 7 Å resolution is overestimated, and the map density should be lowpass filtered to a lower resolution (maybe 15 Å?) for more accurate display. I don't think this will negatively affect the point that the authors are trying to make here.

Line 195, Results: "We did however observe many shell-attached densities in the tomograms, which are best visible in tomograms of broken CB shells (Figure 5D). Some of these densities are the correct size for carbonic anhydrase, which also is capable of binding Rubisco and was previously found to be shell associated"

-> Correct me if I am wrong, but didn't Blikstad et al. 2021 find that the carbonic anhydrase binds Rubisco and not the shell? I quote from that study: "This screen showed that CsoSCA interacted with Rubisco, while none of the other carboxysome proteins had detectable binding above background." Could these shell-attached densities instead be aggregates of CsoS2, which does interact with the shell? Note that rupture of carboxysomes and resulting exposure of CsoS2 to the buffer would likely result in aggregation.

Line 202, Results: "roughly 5% of CBs also contain a large enzyme that is consistent in size and shape with a 20S proteasome (Extended Data Figure 5)."

-> This is very interesting, but it doesn't look like a 20S proteasome to me (I stared at many proteasome tomograms during my postdoc). Without a structural average, it's just guesswork, and I understand that there aren't enough particles for a good average. But... It looks quite similar to this archaeal chaperonin (See Fig. 1 <https://doi.org/10.1016/j.str.2011.03.005>).

Line 218, Discussion: "We observed three Rubisco packing types in the CBs: sparse, dense and ordered (Figure 2A)."

-> For clarity, it would be very helpful to include a small supplemental table detailing the number of carboxysomes in the tomograms that were ruptured / intact sparse / intact dense / intact ordered. This information could also be incorporated into Extended Fig. 1, to help readers understand how parameters such as Rubisco concentration correlate with the above classes.

Line 213, Discussion: "Recently, tomographic analysis of the algal pyrenoid has shown that Rubisco behaves in a liquid-like fashion."

-> The authors could consider additionally citing the recent mechanistic study by He et al. (<https://doi.org/10.1038/s41477-020-00811-y>).

Line 229, Discussion: "Polymerization is a highly effective packing strategy, and does not obstruct the Rubisco active site nor CsoS2 and CsoSCA binding sites."

-> Without getting overly speculative, the authors could expand on this idea a bit to discuss what physiological advantages Rubisco fibril formation could bring. Also, do the authors expect other alpha-carboxysomes to form Rubisco fibrils? What about beta-carboxysomes, which have also not been studied yet at high resolution?

Line 482, Methods: "Rubisco subtomogram average unfiltered half-maps and sharpened full map will be deposited in the EMDB at time of publication. Raw tomogram movie frames may be accessed through the Caltech Electron Tomography Database (<https://etdb.caltech.edu/>) upon publication."

-> Please also deposit example tomograms in the EMDB and raw tomogram movie frames in EMPIAR. The Caltech ETDB is great, but EMDB/EMPIAR remains the main community repository for now, so it should also receive copies of the data.

REVIEWER COMMENTS

Reviewer #1 (Remarks to the Author):

I very much enjoyed reading this manuscript, which describes an interesting set of data relating to packing arrangements of Rubisco in *Halothiobacillus neapolitanus* carboxysomes. It utilizes cryo electron tomography, coupled with existing Rubisco crystal structures, to solve Rubisco arrangements within purified carboxysomes. A take-home result is that high concentrations of Rubisco within carboxysomes leads to increasingly ordered packing of the enzyme, via observable interaction interfaces. While the manuscript provides very interesting insights into Rubisco ordering in these microcompartments, I felt that more could be done to extend the impact of the manuscript and further elucidate the observed interactions between Rubisco holoenzymes. For example, can the interactions be eliminated through mutation of the interacting interfaces? Are the interacting interfaces highly conserved? What would be expected in carboxysomes from different organisms given the conservation (or lack thereof)?

A general question relating to the observed 'classes' of carboxysomes based on their relative packing density (sparse, dense, ordered): could the observed heterogeneity in cbx composition and structure be due, in part, to the growth conditions of Halo and/or the potential lack of growth dependency on cbx rubisco in these organisms (they can grow heterotrophically and have more than one Rubisco). Would a system or a growth condition which enforces cbx growth dependency provide more definitive results? Also, is there a way of confirming that the observed interactions are not artefactual? It is worth noting that similar tightly packed Rubisco, appearing as a paracrystalline array, have been observed in beta carboxysomes (e.g. Kaneko et al.) and in older EM images of pyrenoids, both have which have since been described as liquid condensates which can become even more freely moving within oxidised beta carboxysomes, suggesting that these previous paracrystalline observations were potentially artefactual. Can the authors provide comment to assure the reader the structures observed here are a genuine reflection of functional alpha-cbx?

In general, I think the work is highly interesting but would be suitable for this journal if additional analyses were carried out, as highlighted in specific comments below. I would be extremely happy to review a version of the manuscript with amendments.

We thank the reviewer for this thoughtful review and consideration of biological context. We have added several components to the manuscript in response, including analyses of conservation, packing, and in vivo tomography.

*Our analysis widely differs from the referenced work in other systems. The resolution of our dataset leaves little ambiguity regarding the organization of the CBs in vitro, thanks to our care to accurately identify 98-99% of Rubisco complexes inside the carboxysomes prior to averaging. To guard against potential purification artefacts, we have now confirmed with in vivo tomography that both the dense and ordered CB packing types are present inside *Halothiobacillus neapolitanus* cells. We do believe that the sparse CBs are either over-represented due to our purification protocol or the result of Rubisco leakage from dense CBs (under-represented in vitro compared to in vivo), which we have adjusted the manuscript to clarify.*

The Rubisco-Rubisco binding site does not present clear residues that would direct binding, so we performed a conservation analysis. We found that this site has some of the lowest conservation in the Rubisco complex. While disappointing from the perspective of presenting targets for mutagenesis, this is consistent with our envisioned model of the lattice being

primarily a harm-reduction mechanism (avoiding rampant cytosolic polymerization, tightly-packed crystallization that could obstruct the active site, etc.). A harm-reduction mechanism would result in negative selection that is structurally diffuse, as we see here, rather than the positive selection that would result in clear targets for mutation.

*Given this lack of a single clear target, unfortunately we do not think mutagenesis is feasible in the timescale and scope of this revision. Mutagenesis would require genome modification, and to our knowledge *Halothiobacillus* cannot grow heterotrophically. While Halos do possess an additional Form II rubisco, this can only sustain growth under conditions of elevated CO₂ (>5%). It is known from various knockout experiments that elimination of the carboxysomal rubisco, knockouts abolishing the carboxysome, or even compromising the integrity of the shell lead to high CO₂ requiring phenotypes (Baker et al., *J Bacteriology* 1998; Cai et al., *Life* 2015; Cai et al., *PloS ONE* 2009; Desmarais et al. *Nat. Microbiol.* 2019). Therefore, mutation to the carboxysomal rubisco can have far-reaching effects on the cell that could also affect carboxysomes and the CCM in ways we cannot easily characterize, beyond the obvious potential effects on activity. For all these reasons, rather than attempting an undirected mutagenesis screen we have answered the reviewer's questions with additional wild-type analysis.*

Specific comments:

Line 36: An additional reference for consideration here is the recent review by Hennacy and Jonikas (Annual Review of Plant Biology), further highlighting the importance of transplanting such systems to a non-native host.

We have added this reference. Thank you for pointing it out to us!

Line 40-41: the alpha-cbx of Halo is a 'representative' of cbx's found also in cyano's and other chemoautotrophs?

*Thank you for noticing this. We have adjusted the text to read "The α -carboxysome (CB) is a microcompartment responsible for carbon fixation in many cyanobacteria and chemoautotrophs", and moved our reference to *H. neapolitanus* to the last paragraph of the introduction.*

Line 44: 'could potentially', is the evidence for the Rubisco-S2 interaction inconclusive?

We have adjusted the language to read "A third abundant component, CsoS2, is a disordered scaffold protein that binds both the shell and Rubisco and is responsible for rubisco's encapsulation." The evidence for the interaction is strong, but to our knowledge these clusters have not definitively been proven to contain CsoS2 nor have other contributors been conclusively ruled out.

Fig 1. Gene names should be italicized but not capitalized. Alternatively, protein names should be capitalized and not italicized.

We have adjusted the figure.

Line 56: 'non-equilibrium bicarbonate pool'. This is not clear for those who do not understand CO₂ chemistry. Perhaps expand to indicate that there is specifically a disequilibrium between CO₂ and bicarbonate, in favour of bicarbonate, within the cytoplasm?

We have adjusted this language and expanded the paragraph as suggested. The modified text now reads: "To circumvent these problems, some autotrophic bacteria employ a CO₂ concentrating mechanism (CCM). In the CCM, bicarbonate is actively pumped into the cytosol and then diffuses through the semi-permeable protein shell into the carboxysome where carbonic anhydrase converts it to CO₂. In this way, high local CO₂ concentrations are provided to the encapsulated Rubisco, maximizing its turnover and outcompeting oxygenase activity."

Line 58-61: Models including Kaplan, Badger, Mangan, Long etc. seem to suggest that encapsulation is sufficient. However, all these models are based on unknown diffusional limitations of the shell. Therefore, the primary unknown is the real diffusional resistance of the shell and does not appear to necessitate allosteric regulation of kinetics.

We fully agree with the reviewer that these models suggesting encapsulation as being sufficient also make strong assumptions about shell properties. Yet, the biochemistry of CsoSCA – which is thought to be post-translationally regulated – suggests that not all mechanisms are captured in the current models. We believe that one critical missing item from this is enzyme scaffolding and/or positioning inside the CB, which could compensate for a shell with less diffusional resistance than is generally modeled. However, recent results in the biological condensates literature (e.g. Michael Rosen and colleagues 2021) suggest that enzyme scaffolding can also have allosteric effects making us believe it is important not to rule out this possibility. To begin to study this in a structural fashion, we attempted a comparison of Rubisco fibrils vs. monomeric complexes, but the resolution (7 Å) was not sufficient to rule out the small conformational changes typically associated with allostery. This entire paragraph has been extensively revised, and we have changed the mention of allostery to "other regulation".

Line 66-67: What evidence is presented to suggest this unique feature leads to the rate of CO₂ fixation? Comparison of active site turnover numbers for the different packing classes of carboxysomes would provide this information. Purification of numerous carboxysome preparations from cells grown under a variety of conditions may lead to different proportions of packing types, therefore enabling such analysis. Correlation of packing type and site-dependent Rubisco turnover rates would clarify if packing arrangement led to differential (and possibly allosteric) regulation of Rubisco function.

We have adjusted the language describing the effect of lattice formation to read, "This feature of the alpha-carboxysome may facilitate high-density encapsulation of Rubisco without compromising enzyme activity."

Unfortunately purification appears to have an effect on the proportion of sparse/dense/ordered CBs, which makes a robust in vitro biochemical analysis of packing effects impossible. The liquidity of Rubisco within the CB would also make it difficult to draw conclusions from such an experiment.

Fig 3. A 'fibril' needs defining more clearly. I also don't fully appreciate sub-fig B. Could it be superimposed on a tomogram image for context?

We have heavily adjusted the text of the preceding paragraph that introduces the ordered/dense/sparse classification to include this definition. Unfortunately we cannot overlay the 3B rendering directly onto a tomogram - the tomogram must be displayed as a slice or with strong depth shading to be decipherable, which prevents visualization of the full lattice. Instead, we have changed this panel in several ways. 1) Included a side view of the same rendering, which matches the orientation of the lattice in Figure 2, and indicated the relationship between

the Figure 2 orthoslice and the Figure 3 renderings in the text. 2) Added an inset with a to-scale rendering of the Rubisco zoomed in on one lattice layer. 3) Changed the background of Figure 3B to white, to better bring out the non-lattice Rubiscos. 4) Included Extended Movie 1, which tilts this representation to better bring out the packing. We hope these changes will better contextualize Figure 3B.

Fig 4. Is the observed interaction between Rubisco's (Fig 4B) consistent with observations relating to any known Rubisco mutations, or have such mutations been assessed? Attempts to address this would make for more impactful results, as would some assessment of whether the observed interaction affects Rubisco catalysis.

Fig 5. This observation is highly relevant to observations of rigid rod-like carboxysomes from Halo (Cai et al., csoS4 deletions) and in the chloroplast expressed carboxysomes of Long et al. where it is possible that only shell-bound Rubisco's are present, and they appear to have a specific arrangement. How do the observations here compare? Extended Fig 4 shows the 16 Angstrom offset which may correlate very well with observed packing in rods, possibly contributing to rod-formation by an iterative extension of shell formation and Rubisco organization. Again, this highlights an opportunity to assess this via mutational analysis to manipulate the Rubisco interacting interfaces. In addition, how conserved are the proposed interacting interface residues? It would be safe to assume that there should be a high degree of conservation in these regions of Rubisco large and small subunits.

We agree with the reviewer that the connection between sequence evolution and fibril formation is highly interesting! We have performed an analysis of conservation in Rubisco, with attention to the Rubisco-Rubisco interface (Ext. Fig. 5). Briefly, we find that the small subunit interaction site is one of the least conserved areas of the Rubisco complex. The interaction region does not have a clear set of residues that would be physically responsible for this interaction (Ext. Fig. 5), making it difficult to perform a meaningful mutational analysis in the absence of sequence conservation. The large standard deviation of our twist angle also suggests that, while our pictured interaction is dominant, elimination of it would simply result in a subtle shift rather than a significant disruption of the lattice, which would be difficult to identify outside another full high-resolution analysis (~1.5 years of effort).

We believe that this is an example of a known tendency of D4 proteins to polymerize at high concentrations based on the non-specific interaction of a small number of charged residues coupled with symmetry effects (Garcia-Seisdedos et al, Nature 2017). Based on this principle and the results of this study, a mutational analysis would be inconclusive because the interaction would simply shift to use other charged residues in the vicinity. This is likely the reason why multiple twist angles of the rubisco complexes are found in both our data and in the crystal structures.

*We do observe occasional carboxysomes in vivo that have an elongated morphology (new Figure 6D); unfortunately these do not survive purification. The elongated CBs have Rubisco in layers, which can be either polymerized or not. We also do not observe either direct Rubisco-shell interaction or a set orientation with Rubisco relative to the shell (Figure 5B), which is consistent with the likely tethering of Rubisco to the shell by CsoS2 (a disordered protein that would not be expected to give a strong angular preference). Carboxysomes display a wide range of heterogeneity both within *H. neapolitanus* and in other bacterial species, and there is not currently sufficient understanding of packing morphologies to enable targeted mutagenesis to alter it. In work beyond this paper, we intend to investigate these questions with cellular tomography and hope to have a more satisfying answer in a future publication.*

Fig 5D. Please identify what is assumed to be CsoSCA and Rubisco in this image, as pointed out in line 195-198.

We have added these indicators. The CsoSCA is not based on structure but rather consistent appearance with expected size and shape, so we have labeled it as "unidentified protein density" in the figure legend and included a text description of the possible proteins that could be consistent with this size and shape.

Lines 122-129: This short paragraph seems to take some time to let the reader know that S2 is possibly the link to Rubisco here. Perhaps it could be written more succinctly and definitively. *We have rewritten this paragraph to focus less on the technical reason for not resolving the density, and to instead describe the uncertainty in identifying the binding partner. The paragraph now reads: "Despite this order, no rigid scaffold is observed holding the Rubisco lattice together. An average of adjacent fibrils shows faint densities between the large and small subunits of adjacent Rubiscos (Fig. 3D), suggesting a flexible, sub-stoichiometric binding partner may be present in the lattice. CsoS2 and carbonic anhydrase are both known to bind Rubisco through low-affinity interactions with disordered peptides. CsoS2, in particular, has the appropriate disorder, length and Rubisco binding sites to serve this role. The Rubisco termini have also been suggested to participate in intermolecular interaction, which could provide an alternative route for assembly."*

Lines 174-180: Is there an observed Rubisco:CsoS1 interface? Would observation of rod-like cbx's offer any clues?

We do not observe a direct interface between Rubisco and the shell in any tomograms in vitro or in vivo, including in elongated morphology CBs in vivo. This is consistent with previously published in vivo studies by the Jensen lab and biochemical data from the Savage lab (Oltrogge et al. 2020). We suspect that there is an indirect interaction via a linker protein, most likely CsoS2, which would be consistent with the area adjacent the shell always being occupied first.

Line 202-203: a large 'enzyme' consistent in size and shape with a 20S proteasome? No evidence is presented for what appears to be a disordered aggregate of protein, I assume in the first image in Ext. Fig 5.

The enzyme refers to Ext. Fig. 5B-C; and we have adjusted this language to indicate a "large protein complex".

Ext. Fig. 5A is the aggregate that we refer to in these lines of text; we have adjusted the figure reference to specify 5A and 5B-C for the different clauses in the sentence. The aggregate is a feature occasionally observed but with no known function or identity. Similar dense spots are found in vivo, but have not been assigned a protein identity or role (Iancu ... Jensen, 2010 JMB). CsoS2 would make the most sense as a disordered protein component known to be present at high concentrations in the carboxysome.

Line 223-225: Note that b-cbx have been observed as paracrystalline in cyanobacteria (Kaneko et al, via TEM), despite current knowledge that the interior of these cbx's should be more motile. This raises the question of whether the observations here and in many circumstances might result as artefacts from the sample preparation process or if they are indeed real. This is where further analysis could be extremely helpful. For example, can specific Rubisco mutations still

enable entrapment of the enzyme in carboxysomes but eliminate organised structure at high internal concentrations?

We agree that care needs to be taken to avoid the same controversy as happened in beta carboxysomes. We have answered the question of sample preparation artefacts by collecting an additional in vivo dataset, which confirms that the ordered CBs are found in vivo (Figure 6, Extended Data Table 1, Ext. Fig. 6E).

Importantly, the loose lattice we observe is non-crystalline, and the ordered phase always coexists with at least some Rubisco complexes that do not participate and would remain motile. We believe the lattice retains the ability to reorganize due to the low Rubisco-Rubisco affinity that allows the long fibrils to break easily unless held in place by neighboring fibrils, which was recently echoed in a preprint from the Zhang lab showing that packing morphology can be changed by extreme non-physiological concentrations of calcium. Our second order tensor analysis (Figure 3) is borrowed from liquid crystal theory, which is likely the best way to describe the ordered phase. We have added a sentence in the discussion to clarify this (2nd paragraph): "It is likely that the CB interior can reorganize, given the wobbly Rubisco-Rubisco interaction (visible in the non-zero bending angle, which requires at least 2 of the 4 small subunit interaction sites to be unbound)."

Line 246-247 is a key outcome here. It implies that an evolved, preferential high-concentration-dependent packing will still maintain Rubisco function under 'saturating' carboxysome filling conditions. This should be emphasized, and it implies a calculable upper limit of Rubisco concentrations within carboxysomes which is driven entirely by the observed arrangement.

How does this observed Rubisco packing compare with what has been described as 'Kepler packing' as an estimate of maximum Rubisco quantities in both alpha and beta carboxysomes?

This was a very interesting question. Because the carboxysome lattice is twisted, it cannot tile space indefinitely to create a unit cell definition, so we performed an analysis of occupied space in a 6-fibril area (the number of fibrils that we use to separate dense and ordered morphologies). This analysis gave an upper bound (Kepler packing) for Rubisco density equivalent to roughly 1.1 mM holoenzyme, while our calculated Rubisco concentrations in the CB go up to roughly 900 μ M. We have added this analysis to the paper text. For review only, we also include the below results confirming that the coarse-grain model from Kepler packing is in strong agreement with the atomistic analysis we performed.

Fig. R1

Extended Fig 1. Is it possible to delineate, on these data plots, the apparent 'classes' of cbx (dense, sparse, ordered)?

We have made this adjustment and turned all the Ext. Fig. 1 plots into stacked histograms.

Reviewer #2 (Remarks to the Author):

“Rubisco forms a lattice inside alpha-carboxysomes”
Reviewed by Manon Demulder and Benjamin Engel

Metskias et al. use cryo-electron tomography (cryo-ET) to provide the first high-resolution visualization of the alpha-carboxysome, which has remained rather enigmatic compared to the better understood beta-carboxysome. Surprisingly, the authors observe linear fibrils of Rubisco within some of the isolated carboxysomes. They then generate a subtomogram average of this Rubisco at a fairly impressive 4.5 Å resolution to show that fibril formation is apparently mediated by interactions between the Rubisco small subunits. The polymerization of Rubisco into fibrils is a novel and interesting result, which is of high interest to the whole field of carbon concentration/fixation. The transition from Rubisco liquid condensate into ordered fibril is also of broad interest to the field of phase separation, and may even carry potential interest all the way out to fields such as neurodegeneration, which study these liquid-to-solid state transitions. However, there is one major issue with this study, which should either be addressed with additional experiments, or be acknowledged with careful rewording of the text and its interpretations.

Major issue:

A major limitation of this study is that the cryo-ET was performed on isolated carboxysomes. Some features such as the Rubisco fibrils are likely also present within the cell, as I cannot imagine how isolation of carboxysomes would induce fibril formation. However, the isolation procedure and subsequent blotting onto EM grids are unquestionably disruptive-- this is clearly shown by the aggregate of broken carboxysome shells in Fig. 5D. Furthermore, I know from my own experience that larger isolated structures (e.g., centrioles) are very easily compressed when they are blotted and frozen into a thin layer of ice on an EM grid. This compression is unavoidable, and is often dramatic. Compression would almost certainly impact the organization of Rubisco within the isolated carboxysomes, especially given the flexibility/disorder of the CsoS2 linker. It could also detach Rubisco in the carboxysome interior from the layer of Rubisco more tightly bound to the carboxysome wall. The authors conclude that “the most consistent feature of the CBs is their compositional and structural heterogeneity” (Line 200), but with the present data, it is not possible to say whether this is a result of carboxysome biology or an artifact of the sample preparation. Specifically, the isolation and blotting of carboxysomes could potentially affect the following analysis presented in this study:

*The reviewers bring up valid points about the limitations of studying a purified system. We agree with the reviewers that the most likely artefact is a decrease of order and potentially leakage of Rubisco, rather than the lattice which is the primary finding of the paper. To address this issue we have now recorded cryo-tomograms of intact *H. neapolitanus* cells and inspected the arrangement of Rubisco within carboxysomes in vivo. Again we observed clear Rubisco fibrils, as well as less-ordered but still densely-packed arrangements (see new Fig. 6 and Ext. Fig. 6E).*

In addition, we note that unlike the beta-carboxysome and other complex Rubisco-filled organelles, the alpha-carboxysome has a rich history of purification and in vitro function; most of our biochemical understanding of the alpha-carboxysome is derived from purified microcompartments, so purification passes the most important test of structural biology (continued function of the sample). Compression forces from blotting are indeed a concern but one that is manageable, as evidenced by the history of structural virology in cryo-EM which are

samples of similar size scale and arguably more mechanically delicate. We have adjusted the text to account for these potential effects as well as emphasize the functional nature of purified alpha-carboxysomes.

*Further, other in vivo datasets at lower resolution may be found in previous Jensen lab publications, and are qualitatively consistent with our findings. Manual observation of orthoslices in Iancu et al. and Jensen, JMB 2010 is consistent with the Rubisco distance from the shell and lack of defined angle relative to it (the Rubisco-shell distance is not discussed in our manuscript, but is observable in the orthoslices throughout the figures). We have adjusted the manuscript text to include better references to published in vivo datasets from *H. neapolitanus*.*

1) The relative abundance of Rubisco fibrils (Figs. 2A, 4A)

Line 94: "Roughly one third of the carboxysomes displayed ordered packing"; Line 131: "The ratio of bound and free Rubisco concentrations scales with Rubisco concentration in the CB (Figure 4A)."

-> Are these numbers representative of the biology or the sample preparation?

We have adjusted the text to indicate that this is in vitro, and that the sparse packing may partially result from purification. We have also added in vivo numbers for the morphology comparison (Extended Data Table 1).

2) The bending angle and twist between Rubisco complexes along a fibril (Fig. 4C)

Line 140: "There is a bending angle of 3 ± 2.5 degrees and a 14-degree standard deviation in the twist between stacked Rubiscos."

-> Is this fibril bending and twisting present in the cell, or is it an artifact of compression forces caused by the blotting?

3) The higher-order packing of fibrils into a twisted hexagonal lattice (Fig. 3)

Line 117: "Rendering all Rubisco fibrils within a CB displays the ordered phase: a loose hexagonal lattice-like ultrastructure of Rubisco fibrils twisting about each other with six-fold pseudo-symmetry (Figure 3B). The fibrils are held at a distance of 12.5 ± 0.7 nm with a tilt of 10 ± 3 degrees (Figure 3D)."

-> Again, is this lattice spacing and tilt affected by the blotting forces?

The orientation of the fibrils relative to the compression forces between the air-water interfaces is random, so it's physically unlikely to drive a specific bending angle or twist that is consistent among all these orientations. We have added this to the text of the results (in vitro reference) and methods (relative angle of fibrils to compression and shearing force vectors, included in a new paragraph of potential sample preparation effects relative to data analysis).

The most likely artefact caused by an aligned force on randomly oriented samples would be to increase the noise of the measurement. We have adjusted the results text to reflect this is in vitro, and the new methods text expands on the point that purification and/or blotting may increase the noise in this analysis.

4) The packing and orientations of non-fibril Rubisco in carboxysomes (Extended Figs. 1, 3)

Line 95: "Other CBs displayed a range of Rubisco concentrations, from sparse to dense (Extended Data Figure 1). A nearest-neighbor Rubisco alignment search across all CBs revealed a concentration dependence: Rubisco orientation is random at low concentration (sparse CBs), but becomes increasingly non-random as concentration rises (dense CBs, Extended Data Figure 3)."; Line 205: "Finally, even in the most ordered alpha-carboxysomes, a

substantial portion of the Rubisco does not participate in the lattice (Figure 3B). This heterogeneity is consistent with models of CB assembly, which are based on phase separation and binding affinities rather than a tightly ordered, regular assembly mechanism”

-> Is the observed heterogeneity in Rubisco concentration and organization representative of biological variability or artifacts from the isolation and blotting?

We have confirmed our findings with in vivo data; please see the revised manuscript. Spatial heterogeneity in Rubisco packing is present in previous in vivo studies by the Jensen and Chiu labs on multiple bacterial species and is the most likely reason that the lattice went unnoticed before now. We have adjusted the text near original line 205 to reflect the in vitro nature of our results and their consistency with previous publications. Rubisco concentration (original line 95) has already been addressed above and the text has been adjusted accordingly there - and in the methods - to reflect the possible effect of in vitro study.

5) Rubisco organization and concentration next to the carboxysome shell compared to the carboxysome interior (Fig. 5)

Line 174: “The Rubiscos adjacent to the shell occasionally participate in the long fibrils, but not consistently, and appear to behave differently than Rubisco in the CB interior.”; Line 176: “the layer is well-populated even in otherwise sparse CBs. We analyzed the angle of the Rubisco C4 axis relative to the shell and found a predominantly random distribution... (Figure 5B).”

-> Is this distinction in the organization of shell-adjacent Rubisco also seen within native cells, or have forces from the isolation and blotting detached Rubisco fibrils in the carboxysome interior from the layer of Rubisco more tightly bound to the carboxysome wall?

The lack of defined angle relative to the shell, and the inconsistent participation in lattices, are present in vivo as well. Please see the new Figure 6 and Ext. Fig. 6E for orthoslices. This is also consistent with current models of alpha-carboxysome biogenesis.

6) Proposals about carboxysome assembly/maturation mechanisms (Discussion)

Line 219: “Sparse carboxysomes lack the concentration necessary to form an ultrastructure, but the dense and ordered packings overlap in Rubisco concentration. All packing types display polymerization of the Rubisco; therefore, the primary distinction between dense and ordered CB packing is whether the fibrils align”; Line 226: “Therefore, the phase transition between dense and ordered packing may involve the CB maturation trajectory.”

-> Are we really seeing a maturation-driven phase transition, or are some carboxysomes more disrupted than others?

Existing in vivo datasets show a lack of paracrystalline order in the alpha-carboxysome (see cited works from the Jensen and Chiu labs), so a spectrum of order is consistent with these data. This is also found in our in vivo data, which show both dense and ordered CBs inside cells. In our response to comment 4 above we added text on in vivo heterogeneity that will also address this.

Line 219 is simply a description of our classification and a reminder to the reader that all polymerization is inherently concentration-dependent and thus a concentration threshold to Rubisco ultrastructure would be expected on a physical basis. Line 226 was speculative and we have replaced it with, "but the relationship between spatial heterogeneity, mobility and chemical environment remains unproven in this system".

I am surprised and a bit disappointed that the authors did not complement this analysis of isolated carboxysomes with some bona fide in situ data of FIB-milled *Halothiobacillus* cells. This

approach is becoming mainstream for leading cryo-ET labs, and indeed, the last author of this study has access to the necessary FIB/SEM instrumentation, as evidenced by their recent publications:

Swulius et al. 2018 <https://doi.org/10.1073/pnas.1711218115> ;

Martynowycz et al. 2019 <https://doi.org/10.1016/j.str.2018.12.003> ;

Zhang et al. 2020 <https://doi.org/10.1126/sciadv.abc8258> ;

Carter et al. 2020 <https://doi.org/10.1126/sciadv.aay9572> ;

Mageswaran et al. 2021 <https://doi.org/10.1101/2021.12.13.472487> ;

Nicolas et al. 2022 <https://doi.org/10.1101/2022.01.31.478342> ;

Regardless of issues associated with the isolation procedure, the Rubisco fibrils appear to be real, and they are a novel finding of high interest to the field. As far as I can tell, the subtomogram averaging performed on these fibrils is methodologically solid, providing some clues into how the fibrils may form via interactions between the small subunits. Thus, I see two possible routes to publication: 1) The authors explicitly describe this work as an in vitro study of isolated carboxysomes. They emphasize the aspects that likely remain true in these isolated conditions, such as the Rubisco-Rubisco structural interactions that establish the fibrils, while the other analyses that may be affected by the isolation (outlined above) should be carefully qualified. 2) The authors add complementary in situ cryo-ET from FIB-milled cells. This does not have to be an extensive dataset, but in situ analysis would be highly valuable to confirm the in vitro observations and add reliable information about the prevalence of different Rubisco organizations within native cells. I think route #2 is definitely preferred and will significantly increase the impact of this study. However, if the authors opt for route #1, I would not object to publication as long as the conclusions are properly qualified.

We have now both added in vivo cryo-ET data and adjusted the text for purified CBs. We agree with the Reviewers that the original results reported in the reviewed draft were not in vivo and have clarified that language throughout, and we have now highlighted the possible problems associated with purification and discussed them. We have also now recorded tilt-series of intact cells to confirm all the major findings (shown now in the new Figure 6 and Ext. Fig. 6E). While we agree that a FIB-milled dataset would be ideal, one of sufficient quality and quantity to address the reviewers' concerns as to tilt, twist, etc. is not feasible for us in the timescale and context of this revision.

Minor issues and requested text changes:

Line 22, Abstract: "Here, we use cryo electron tomography and subtomogram averaging to determine an in situ structure of Rubisco at 4.5 Å"

-> This is certainly NOT an in situ structure. In situ means within the original natural environment. Specifically regarding cryo-ET, in situ means inside the cell, as has been demonstrated in many previous cryo-ET studies of intact small cells or larger cells that have been thinned with a cryo-microtome or by focused ion beam milling. The Rubisco subtomogram average in this study is derived from isolated carboxysomes, which are not in their native state.

We have adjusted the text to read "a 4.5 Å structure of Rubisco inside the alpha-carboxysomes".

Line 29, Abstract: In the abstract, it is claimed that the insights into Rubisco fibril formation "provide future directions for bioengineering of microcompartments."

-> In the discussion section, the authors should provide some detail on how Rubisco fibrils open

these new bioengineering opportunities. Conversely, if it is too early to predict these possible applications, this claim could (should!) be removed from the abstract.

We have added this to the discussion (final sentence, 3rd paragraph from the end). It's likely that including a weak-affinity polymerization in cargo enzymes could assist their incorporation at high concentration. The Rubisco concentrations inside the CBs can approach Kepler packing (see Reviewer 1 response), which is not typically seen in liquid-liquid phase separation alone (the current CsoS2 model in the field).

Line 66, Introduction: "This unique feature of the alpha-carboxysome likely helps to increase the rate of carbon fixation inside the CB."

-> How do the authors know that Rubisco polymerization is a unique feature of alpha-carboxysomes? Beta-carboxysomes also contain a pseudo-crystalline lattice, which has not yet been studied at high resolution. Also, the authors observe different degrees of order with similar Rubisco concentrations, so fibrils aren't simply a mechanism for increasing the concentration of Rubisco within the carboxysome (Line 219: "the dense and ordered packings overlap in Rubisco concentration"). Thus, without mechanistic studies specifically disrupting the fibrils, it is probably too early to conclude that Rubisco polymerization likely helps increase the rate of carbon fixation.

We have replaced the text in question with "This feature of the alpha-carboxysome may facilitate high-density encapsulation of Rubisco without compromising enzyme activity."

Line 117, Results: "Rendering all Rubisco fibrils within a CB displays the ordered phase: a loose hexagonal lattice-like ultrastructure of Rubisco fibrils twisting about each other with six-fold pseudo-symmetry (Figure 3B)."

-> How often was this hexagonal lattice observed in the dataset of 139 carboxysomes? Figs. 3A and 3C appear to analyze many carboxysomes, but the prevalence of this lattice is not clear to me. The figure legend claims that Fig. 3B is a representative example. In the supplement, please show at least four more examples of such hexagonal lattices in other carboxysomes.

We apologize for confusion on this point. The lattice is how we define the "ordered" carboxysomes, so the prevalence is a little less than half the dataset though the size/number of concentric layers varies. We have adjusted the text (paragraph starting at original line 94) to better link the ordered morphology classification with the lattice. We have also included the additional orthoslices in the Extended figure as requested and a table with numbers. In keeping with best practice in biophysics, all analyses are performed on all data, as described in the methods.

Line 119, Results: "The fibrils are held at a distance of 12.5 ± 0.7 nm with a tilt of 10 ± 3 degrees (Figure 3D)."

-> Is the packing and helical tilt between neighbor fibrils always the same, or are there variations between carboxysomes? If not done already, please quantify this for the full dataset (It's not clear to me whether these numbers are averaged from every hexagonal lattice in the dataset or just from the example in Fig. 3B).

This analysis is for the entire dataset, otherwise the data would not be sufficient to justify use of Gaussian distributions in the analysis due to the small number of fibrils per CB. We have adjusted the methods text to better reflect this. We do not observe quantifiable variation between carboxysomes, but it's possible that this is due to the combination of a loosely-ordered lattice and a relatively small number of both dependent and independent datapoints rather than a lack of biological variance between CBs.

Line 124, Results: “ We hypothesize that a disordered, lower-occupancy binding partner may maintain a maximum distance and promote tilt between fibrils. Such a linker would likely not be visible in a subtomogram average due to the combination of low occupancy and disorder, especially if there were also disorder in the binding site, causing the system to act as a 'fuzzy complex'. CsoS2 has the appropriate disorder, length and Rubisco binding sites to serve this role.”

-> Wouldn't a low-occupancy disordered linker promote significant flexibility between the Rubisco fibrils instead of this specific spacing and tilt? Perhaps there is a more physical packing explanation for the parameters of the observed hexagonal lattice?

We have adjusted the text to remove the mention of maximal distance and tilt, as well as other items requested by the other reviewer. Physically, a maximal distance would suffice for our distributions, as the minimal distance is provided by Rubisco itself and Brownian motion would take care of the rest within that confined space. However, we agree that the tilt is more difficult to explain. We have not identified a physical principle that would explain the tilt in the absence of an organizing partner to at least maintain a specific distance between the chains, and a simulation would require parameterization beyond what current knowledge can justify.

Line 165, Results: “Rubisco crystal structures also display variable interactions in this region. H. neapolitanus Rubisco structures 1SVD and 6UEW both show alternative longitudinal interactions in the crystal structure involving helix 3 (Extended Data Figure 4). Both crystal structure interactions lie within the range represented in our data, suggesting these conformations may be sampled in vivo (though not the dominant conformation).”

-> Two comments here. First, please be careful with the wording because isolated carboxysomes are not in vivo. Second, it would be helpful to provide some context for these two PDB structures. 1SVD is an apo structure, whereas 6UEW is bound by CsoS2 N-peptide, but this is not clear from the text.

We have adjusted the text to include context for the crystal structures.

We apologize if this text is not clear. We are stating that the Rubisco-Rubisco interaction inside the crystal structures is also present in purified carboxysomes, and thus is likely not a crystal packing artefact. (Please also note we do not say "representative" but "sampled", which allows for shifts between purified CBs and CBs inside cells. We have added extensive references to the purified nature of the dataset in response to other points in this review, so this will be clear to the reader in the revised manuscript.)

Line 187, Results: “7 Å C1 interior Rubisco subtomogram average” (shown in Fig. 5C)

-> This Rubisco average appears to be filtered to a resolution that is too high, resulting in noisy spikey extensions all over the structure. Likely the 7 Å resolution is overestimated, and the map density should be lowpass filtered to a lower resolution (maybe 15 Å?) for more accurate display. I don't think this will negatively affect the point that the authors are trying to make here.

We agree this map is over-sharpened; it's a known and inherent problem in post-processing C1 subtomogram averages at lower resolutions (for another recent example of this known issue, see Qu ... Briggs, Science 2022). The local resolution in this slab view is also lower than the global resolution due to its location on the exterior of the enzyme (also commonly seen). We have shifted Figure 5C to a simple low-pass filtered average (unsharpened, all data), and removed references to a specific resolution to avoid confusion between local and global.

Line 195, Results: “We did however observe many shell-attached densities in the tomograms, which are best visible in tomograms of broken CB shells (Figure 5D). Some of these densities are the correct size for carbonic anhydrase, which also is capable of binding Rubisco and was previously found to be shell associated”

-> Correct me if I am wrong, but didn't Blikstad et al. 2021 find that the carbonic anhydrase binds Rubisco and not the shell? I quote from that study: “This screen showed that CsoSCA interacted with Rubisco, while none of the other carboxysome proteins had detectable binding above background.” Could these shell-attached densities instead be aggregates of CsoS2, which does interact with the shell? Note that rupture of carboxysomes and resulting exposure of CsoS2 to the buffer would likely result in aggregation.

We have added the possibility of CsoS2 aggregate to the text, as well as the rubisco activase which is probably more likely (IDP aggregation does not typically yield a specific, dense shape of limited molecular weight). While the Blikstad 2021 paper does indeed indicate Rubisco binding rather than shell interaction, the "S" in CsoSCA stands for shell due to earlier papers (cited in the text) indicating historical observations of binding. It's possible that all these papers may be correct, as the PPI data may not perfectly mimic the chemical environment inside the carboxysome (which may also shift over time).

Line 202, Results: “roughly 5% of CBs also contain a large enzyme that is consistent in size and shape with a 20S proteasome (Extended Data Figure 5).”

-> This is very interesting, but it doesn't look like a 20S proteasome to me (I stared at many proteasome tomograms during my postdoc). Without a structural average, it's just guesswork, and I understand that there aren't enough particles for a good average. But... It looks quite similar to this archaeal chaperonin (See Fig. 1 <https://doi.org/10.1016/j.str.2011.03.005>).

Thank you for pointing this out to us! You're right that this looks like a better fit. We have adjusted the text to remove the reference to the 20S proteasome. We did not specifically mention the chaperonin lest we fall into the same trap, but we will continue to look for these and attempt a low-resolution average in the future if we can compile enough particles.

Line 218, Discussion: “We observed three Rubisco packing types in the CBs: sparse, dense and ordered (Figure 2A).”

-> For clarity, it would be very helpful to include a small supplemental table detailing the number of carboxysomes in the tomograms that were ruptured / intact sparse / intact dense / intact ordered. This information could also be incorporated into Extended Fig. 1, to help readers understand how parameters such as Rubisco concentration correlate with the above classes.

We have incorporated this into Ext. Fig. 1 as requested, and added Extended Data Table 1 for both in vitro and in vivo datasets.

Visibly ruptured carboxysomes are occasionally present in the sample but as broken shells (Figure 5D); they are intentionally avoided during data collection so an unbiased estimate cannot be made. Rubisco attached to empty shells was not analyzed. (NB: If the sparse CBs do represent Rubisco leakage/loss, which we agree is possible, the most likely explanation for the lack of visible shell rupture is that the shell reassociates prior to freezing.)

Line 213, Discussion: “Recently, tomographic analysis of the algal pyrenoid has shown that Rubisco behaves in a liquid-like fashion.”

-> The authors could consider additionally citing the recent mechanistic study by He et al. (<https://doi.org/10.1038/s41477-020-00811-y>).

We have incorporated this citation, thank you for pointing it out to us.

Line 229, Discussion: "Polymerization is a highly effective packing strategy, and does not obstruct the Rubisco active site nor CsoS2 and CsoSCA binding sites."

-> Without getting overly speculative, the authors could expand on this idea a bit to discuss what physiological advantages Rubisco fibril formation could bring. Also, do the authors expect other alpha-carboxysomes to form Rubisco fibrils? What about beta-carboxysomes, which have also not been studied yet at high resolution?

*We have reorganized this part of the discussion to better emphasize the advantages while avoiding too much speculation as we cannot knock out the polymerization. We believe this is a "do no harm" mechanism for maximizing Rubisco concentration without sterically choking the enzyme. Other systems may not need this mechanism, as it may be specific to alpha-carboxysome assembly where the Rubisco condenses in the cytoplasm at near-maximal concentrations. Because the *H. neapolitanus* Rubisco forms polymers in crystal structures, we examined other Rubisco crystal structures for these packing interactions. This phenomenon seems to be specific to Form 1A Rubisco.*

Beta-carboxysomes are quite interesting. It's important to note that liquid crystal or paracrystalline packing does not necessitate fibril formation; they may indeed have this, but without a fibril-containing crystal structure or a resolved interaction site in situ it would be too speculative to discuss here.

Line 482, Methods: "Rubisco subtomogram average unfiltered half-maps and sharpened full map will be deposited in the EMDB at time of publication. Raw tomogram movie frames may be accessed through the Caltech Electron Tomography Database (<https://etdb.caltech.edu/>) upon publication."

-> Please also deposit example tomograms in the EMDB and raw tomogram movie frames in EMPIAR. The Caltech ETDB is great, but EMDB/EMPIAR remains the main community repository for now, so it should also receive copies of the data.

We will additionally deposit example tomograms and frames in EMDB/EMPIAR as requested. This paragraph will be updated with accession numbers and other pertinent information during final preparation of an accepted manuscript.

Reviewers' Comments:

Reviewer #1:

Remarks to the Author:

It was a pleasure to read the ammended manuscript and the meaningful responses from the authors to both reviews. The responses are well considered and, from this reviewer's viewpoint, well-answered with reasonable action taken to address issues raised. I fail to find any outstanding issues that would not be best answered by the greater community rather than another round of review. I therefore recommend publication of the manuscript in its current form.

Reviewer #2:

Remarks to the Author:

We read with much interest the new additions to the manuscript and were glad to see that a few tomograms were added to illustrate the in-situ relevance of the study. We are also satisfied with the nuances added in the text to clarify in situ vs in vitro, as well as the other amendments to the text.

One minor suggestion for line 275, would be that the authors specify what Keppler packing is and why it matters in this context. Currently this is only very briefly specified in the materials and methods

REVIEWER COMMENTS

Reviewer #1 (Remarks to the Author):

It was a pleasure to read the ammended manuscript and the meaningful responses from the authors to both reviews. The responses are well considered and, from this reviewer's viewpoint, well-answered with reasonable action taken to address issues raised. I fail to find any outstanding issues that would not be best answered by the greater community rather than another round of review. I therefore recommend publication of the manuscript in its current form.

Thank you for the time you spent on your review for us! We appreciate the thoughtful comments and improvements they led us to make to the paper.

Reviewer #2 (Remarks to the Author):

We read with much interest the new additions to the manuscript and were glad to see that a few tomograms were added to illustrate the in-situ relevance of the study. We are also satisfied with the nuances added in the text to clarify in situ vs in vitro, as well as the other amendments to the text.

Thank you for the time you spent on your review for us! We look forward to following up on many points in subsequent publications.

One minor suggestion for line 275, would be that the authors specify what Kepler packing is and why it matters in this context. Currently this is only very briefly specified in the materials and methods

We have added a definition of Kepler packing to this sentence that also suggests its relevance: "Rubisco polymerization may also assist in CsoS2-based Rubisco condensation during CB shell assembly, facilitating its incorporation at concentrations that can exceed some crystal structures and approach Kepler packing (the maximal density for packing of spherical objects)."